# Visual Prompting Reimagined: The Power of Activation Prompts

## Abstract

Visual prompting (VP) has emerged as a popular method to repurpose large pretrained models for downstream vision tasks. Unlike many parameter-efficient fine-tuning (PEFT) techniques that modify model parameters, VP introduces a universal perturbation directly into the input data to facilitate task-specific fine-tuning while keeping the pretrained model intact. However, there exists a noticeable performance gap between VP and conventional fine-tuning methods, highlighting an unexplored realm in theory and practice to understand and advance VP to close its performance gap. Towards this end, we introduce a novel concept, termed activation prompt (AP), which extends the scope of input-level VP by enabling universal perturbations to be applied to activation maps within the intermediate layers of the model. With the aid of AP, we show that VP, by its input perturbation design, has intrinsic limitations in both performance and efficiency. By contrast, AP shares a natural connection to normalization tuning, *e.g.*, batch normalization for convolutional neural networks (CNNs) and layer normalization for vision transformers (ViTs). This illuminates the reason behind the observed better accuracy of normalization tuning than VP in the literature. Furthermore, we show that the choice of prompting exhibits a distinct preference for layer depth, with conclusions varying significantly between CNNs and ViTs. We theoretically elucidate the rationale behind such preference by analyzing global features across layers. By conducting extensive experiments across 29 datasets and various model architectures, we provide a thorough performance analysis of AP, comparing it with VP and PEFT baselines. Our experimental results demonstrate that AP significantly surpasses the input-level VP in terms of both accuracy and efficiency, considering factors like time, parameters, memory usage, and throughout. These results further support our new insights into the incapabilities of VP and the capabilities of AP.

## 1 Introduction

Large pretrained models have emerged as fundamental components in deep learning (DL) research in recent years (Brown et al., 2020; Touvron et al., 2023; Chiang et al., 2023; Li et al., 2022; Bai et al., 2023a). Thus, the pretraining-finetuning paradigm rises, allowing for quickly adapting a pretrained model to a plethora of downstream tasks (Jia et al., 2022; Hu et al., 2021; Chen et al., 2022b; Cai et al., 2020; Sung et al., 2022; Pfeiffer et al., 2020; Chen et al., 2023b). Due to its extensive parameter space, the model possesses ample capacity to acquire a rich and diverse set of valuable features during pretraining, thereby ensuring exceptional performance and high data efficiency during finetuning. Nonetheless, the substantial increase in computational demands, as highlighted in recent studies (Frantar & Alistarh, 2023), has underlined the need for more economical and lightweight fine-tuning approaches. Consequently, the development of such approaches has become a central research focus.

Among the various parameter-efficient finetuning (PEFT) methods (He et al., 2021; Hu et al., 2021; Pfeiffer et al., 2020; Chen et al., 2022b; Xu et al., 2023), prompting technique has been gaining popularity (Liu et al., 2023; Li & Liang, 2021). Different from the model-centric PEFT techniques in the realm of computer vision (CV), the conventional visual prompting (**VP**) crafts specific prompts or templates on the inputs to reprogram the pretrained model without altering the model parameters, which offers a new data-centric viewpoint to analyze, understand, and further harness the pretrained model (Chen et al., 2023b). However, even though several VP variants have been introduced, current state-of-the-art (SOTA) VP techniques still lag in performance when compared to finetuning-based approaches (Chen et al., 2023b; Wu et al., 2022) and remain in nascent stages, compared to its

counterpart in natural language processing (NLP) (Liu et al., 2023; Li & Liang, 2021), which has found great utility. Therefore, to interpret the root cause for such an inferior performance and to further advance VP, in this work, we primarily focus on the following question:

> **(Q)** *Is VP (visual prompting) truly beneficial for improving vision models and tasks, and under what conditions does it prove effective or ineffective?*

To address the above question, we introduce a more generalized version of VP termed activation prompt (**AP**); see **Fig. 1** for an illustration. In this context, it can be viewed that VP operates as a specific instance of AP, functioning at the input layer rather than any intermediate layer that AP can be applied to. Through a warm-up study of AP in different model layers, we find that VP seems not to be the optimal design in terms of either effectiveness or efficiency. In particular, a properly installed AP can significantly exceed the performance of the conventional input-based VP. To understand the mechanism behind AP, we provide empirical studies as well as theoretical understanding, highlighting a close connection between AP and normalization tuning, although AP does not require modifying model parameters as normalization tuning does.

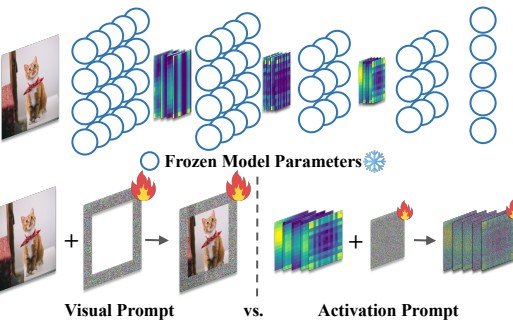

Figure 1: An illustration of the proposed activation prompt vs. the conventional input-based prompt.

Probing deeper into the conditions under which AP can be employed to maximize downstream performance, our analyses shed light on the layer and architectural influences in AP. Specifically, AP has a pronounced layer preference to achieve the best performance. Interestingly, this preference manifests oppositely in widely adopted architectures like the ResNet and the Vision Transformer (ViT) family. Comprehensive examinations of intermediate features from ResNets and ViTs reveal that layers rich in *global* features typically amplify the efficacy of AP. Conclusively, our theoretical analysis rigorously validates the "layer and architecture effect". Our **contributions** are as follows:

• We propose AP (activation prompt) as a valuable tool for gaining insights into VP (visual prompting). In addition, AP establishes itself as a versatile and highly effective prompting technique in its own right, revealing a provable relationship with normalization tuning.

• We show that the choice of the optimal layer for implementing AP is heavily contingent on the specific model architecture employed. Empirical analyses underscore that this layer-architecture effect is intricately linked to the model's capacity to capture global features. Moreover, we provide theoretical validation to substantiate these empirical findings.

• Through extensive experimentation involving 29 datasets across various benchmarks, we confirm that AP markedly improves upon VP in diverse learning scenarios. When compared to 6 other SOTA PEFT methods, we show that AP excels in both accuracy performance and efficiency.

## 2 RELATED WORK

**Visual prompting.** VP was first proposed in (Bahng et al., 2022b) to extend the prompting technique in NLP. A similar idea with a different name, known as adversarial reprogramming, was also proposed earlier in CV (Elsayed et al., 2018; Chen, 2022; Neekhara et al., 2018; 2022; Chen et al., 2021; Zhang et al., 2022a; Chen et al., 2022a), which aims at re-purposing a fixed pretrained model to adapt to a new task. Recent advancement either focuses on improved label mapping (Chen et al., 2021; Yang et al., 2023) or normalization strategy (Wu et al., 2022) to enhance VP. Other works further extend the idea of VP to areas like adversarial defense (Chen et al., 2023a; Mao et al., 2022) and distribution shift (Huang et al., 2023; Tsai et al., 2023), and vision-language models (Zhou et al., 2022). The most relevant work to ours is VPT (Jia et al., 2022), which proposes another form of prompt by appending additional tokens to *all* the intermediate layers of a ViT. However, it fails at dissecting the layer effect of different models and also results in a more computationally intensive design.

**Theoretical study on prompt engineering.** Existing theoretical works on prompt engineering include the expressive power of the introduced parameter (Wei et al., 2021; Bai et al., 2023b; Akyürek et al.,

2022), the optimization process (Ding et al., 2022; Von Oswald et al., 2023), and the generalization analysis (Xie et al., 2021; Oymak et al., 2023; Zhang et al., 2023a; Li et al., 2023b). Most studies concentrate on in-context learning, a tuning-free hard prompt method. In contrast, for soft prompt tuning, Wei et al. (2021) unveils prompting is powerful enough to remove nonessential information for the downstream task. Ding et al. (2022) interprets prompt tuning as a subspace optimization method for the solution or functional space. Notably, (Oymak et al., 2023) is the sole study on the generalization dynamics of gradient-based prompt tuning but relies on a single-layer Transformer architecture without the MLP layer, making it incapable of examining the impact of multiple layers.

**Parameter-efficient finetuning.** PEFT demonstrates that only finetuning a small part of a large pretrained model can achieve outstanding performance. In the domain of CV, besides prompting-based methods, PEFT methods can be roughly classified into two categories. The former (Basu et al., 2023; Xu et al., 2023) focuses on identifying a small ratio of parameters from the pretrained model itself, such as normalization tuning (Basu et al., 2023), and the latter designs small modules to the original network backbone to adapt do downstream tasks (Hu et al., 2021; Chen et al., 2022b; Karimi Mahabadi et al., 2021; Xu et al., 2023; Pfeiffer et al., 2020; Lian et al., 2022; Zhang et al., 2022b; Luo et al., 2023). The representative work includes LoRA (Hu et al., 2021), which inserts low-rank parameters to the attention blocks of ViT, adapter-based methods (Chen et al., 2022b; Luo et al., 2023; Karimi Mahabadi et al., 2021; Pfeiffer et al., 2020) that interpose lightweight networks within the pretrained model, and FACT (Jie & Deng, 2023), that tensorizes the ViT weights to a 3D tensor and reduces the tunable parameter ratio to less than $0.01\%$. Nonetheless, these strategies heavily rely on the model architecture and thus require additional parameters, which either introduce additional inference latency or only apply to a certain type of model. AP differentiates itself from the methods above by not introducing additional inference overheads or having any requirements on the model architectures.

## 3    ACTIVATION PROMPTING: AN "IN-DEPTH" EXTENSION OF VP

**Preliminaries on VP.** The VP technique harnesses universal pixel-level perturbations applied to input images as a means of model adaptation (Bahng et al., 2022a). For example, VP enables the transfer learning of an ImageNet-trained source model to various downstream tasks without the need for fine-tuning the model weights. It has sparked significant interest in the recent research (Bahng et al., 2022a; Chen et al., 2023b; Zhang et al., 2022a; Tsai et al., 2020; Wu et al., 2022). To be concrete, let $f_{\boldsymbol{\theta}}$ denote the pre-trained source model parameterized by $\boldsymbol{\theta}$, and $\mathcal{D} = \{(\boldsymbol{x}_1, y_1), (\boldsymbol{x}_2, y_2), \ldots, (\boldsymbol{x}_N, y_N)\}$ denote the fine-tuning dataset for a downstream task, with $\boldsymbol{x}$ and $y$ being the data feature and label, respectively. **The objective of VP** is to obtain a perturbation vector, denoted as $\boldsymbol{\delta}_{\mathrm{VP}}$, which is tailored to a specific task but remains agnostic to the input data. This vector is then used to transform the input data $\boldsymbol{x}$ through the function $g(\boldsymbol{x}, \boldsymbol{\delta}_{\mathrm{VP}})$. Here $g$ symbolizes the transformation template function that molds the input image to fit the desired prompt pattern. Two prevalent templates include the addition $g(\boldsymbol{x}, \boldsymbol{\delta}_{\mathrm{VP}}) = \boldsymbol{x} + \boldsymbol{\delta}_{\mathrm{VP}}$ (Bahng et al., 2022a; Zhang et al., 2022a), and the resize-and-concatenation $g(\boldsymbol{x}, \boldsymbol{\delta}_{\mathrm{VP}}) = [\boldsymbol{\delta}_{\mathrm{VP}}, M(\boldsymbol{x})]$ (Zhang et al., 2022a; Chen et al., 2023b), where $M$ stands for the resizing function. Unless specified otherwise, the additive VP formulation is adopted in this work.

**Activation prompts: Generalizing VP to the feature space.** The conventional VP approach primarily focuses on making direct modifications to the input data. However, this direct manipulation has limitations in terms of flexibility and efficiency for two key reasons. *First*, raw input data typically contains an abundance of details, which can introduce complications for tasks like prompt generation due to issues such as background clutter and semantic ambiguity (Yu et al., 2017). In contrast, intermediate features tend to encompass a broader range of local and global attributes, preserving more class-discriminative information for decision-making (Bau et al., 2017). *Second*, parameter updates in VP demand gradient propagation throughout the entire network. Consequently, even with a lower number of tunable parameters, there might not be a substantial increase in training efficiency.

Motivated by the above, we broaden the scope of VP into the feature domain and introduce the concept of **activation prompting (AP)**, see Fig. 1 for an illustration. Given a neural network model with $L$ layers, represented as $\boldsymbol{\theta} = [\boldsymbol{\theta}^{(1)}, \boldsymbol{\theta}^{(2)}, \ldots, \boldsymbol{\theta}^{(L)}]$, the output from the $l$-th layer is denoted as $\boldsymbol{z}^{(l)} = f_{\boldsymbol{\theta}^{(l)}}(\boldsymbol{z}^{(l-1)})$, where $\boldsymbol{z}^{(0)} = \boldsymbol{x}$ (*i.e.*, the input date). Similar to VP, AP at the $l$-th layer is defined by a perturbation vector $\boldsymbol{\delta}^{(l)}$ to the intermediate feature $\boldsymbol{z}^{(l)}$, leading to the 'prompted' feature map $g(\boldsymbol{z}^{(l)}, \boldsymbol{\delta}^{(l)}) = \boldsymbol{z}^{(l)} + \boldsymbol{\delta}^{(l)}$. We denote the output with the $l$-th-layer AP given $\boldsymbol{\theta}$ as $f_{\boldsymbol{\theta}}(\boldsymbol{x}, \boldsymbol{\delta}^{(l)})$.

**The objective of AP** is then to optimize $\boldsymbol{\delta}^{(l)}$ so as to facilitate the adaptation of the fixed source model $f_{\boldsymbol{\theta}}$ for performing the downstream task on $\mathcal{D}$. It is evident that AP can be conceptualized as an extension of VP when we set the layer number $l$ to $0$. Moreover, the optimization process for both VP and AP can be carried out similarly through empirical risk minimization (ERM) on $\mathcal{D}$, *i.e.*, $\min_{\boldsymbol{\delta}^{(l)}} \mathbb{E}_{(\boldsymbol{x},y)\sim\mathcal{D}}\ell(f_{\boldsymbol{\theta}}(\boldsymbol{x}, \boldsymbol{\delta}^{(l)}); y)$, where $\ell$ is the sample-wise cross-entropy loss.

However, AP also exhibits several notable attributes different from VP. *First*, the number of parameters in AP directly relates to the size of the feature map $\boldsymbol{z}^{(l)}$. Hence, a properly designed AP can substantially reduce the parameter count. *Second*, while the optimization of AP mirrors that of VP, its parameter update does not necessitate back-propagation throughout the entire network. For example, embedding AP deeper within the architecture reduces computational demands during training.

**AP might be a better design than VP.** Next, we present a preliminary experiment that serves as a warm-up, demonstrating how AP exhibits the potential to improve accuracy performance, as well as enhance computation and parameter efficiency when compared to VP. We examine the commonly used transfer learning scenario for applying VP, in which the source model ResNet-101 (He et al., 2016) is initially trained on ImageNet (Deng et al., 2009) and is subsequently transferred to the CIFAR-10 dataset (Krizhevsky et al., 2009). **Fig.** 2 presents a performance comparison between AP and VP against the layer index on ResNet-101, at which AP is introduced.

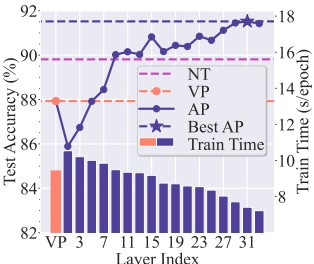

The preliminary results provide several key insights, which will be substantiated in more detail later. *First*, AP holds the potential to substantially enhance the accuracy of transfer learning when compared to VP. For instance, when AP is applied at layer 31, it achieves the highest accuracy in transfer learning, surpassing VP by approximately 5%. In fact, more comprehensive experiments presented in Sec. 5 demonstrate that applying AP to a *deeper* layer consistently produces the most significant accuracy improvements across a wide range of convolutional neural networks (CNNs). *Second*, due to the preference for deeper layers when utilizing AP in CNNs, there exists a computational advantage since back-propagation from the output to the input layer is *not* required. *Third*, AP maintains the parameter efficiency merit compared to VP. For instance, at the layer that exhibits the best performance, AP utilizes only $100k$ parameters, whereas VP employs $150k$ parameters. The results from the

Figure 2: Performance and efficiency comparison of VP, NORM-TUNE and AP over different layer depths of ResNet-101 on OxfordPets.

warm-up experiment above indicate that *AP has the potential to outperform VP, offering not only improved accuracy but also greater efficiency*.

**Understanding AP through its connection to normalization tuning.** In this study, we present normalization tuning (NORM-TUNE) as a parameter-efficient fine-tuning (PEFT) technique. This approach tunes parameters within a model's normalization layers, *i.e.*, BatchNorm for CNNs (Ioffe & Szegedy, 2015) and LayerNorm for ViTs (Ba et al., 2016). For clarity, we denote the tunable parameters of a normalization layer as $\boldsymbol{\gamma} = (\gamma_1, \cdots, \gamma_{D'})^{\top}$ and $\boldsymbol{\beta} = (\beta_1, \cdots, \beta_{D'})^{\top}$, with $D'$ representing the number of channels or the token dimension. Further, define $\boldsymbol{\mu}$ and $\boldsymbol{\sigma}$ as the channel-wise mean and standard deviation constants of $\boldsymbol{z}^{(l)}$ for BatchNorm over the entire batch. For LayerNorm, they represent the data-wise mean and standard deviation of $\boldsymbol{z}^{(l)}$ across the embedding dimension. Given that both AP and NORM-TUNE



Figure 3: Tunable parameter shape comparison between NORM-TUNE and AP (ours). The same color indicates shared parameters across different dimensions.

utilize a linear model for feature representations, namely $g(\boldsymbol{z}^{(l)}, \boldsymbol{\delta}^{(l)}) = \boldsymbol{z}^{(l)} + \boldsymbol{\delta}^{(l)}$ for AP and $g(\boldsymbol{z}^{(l)}, \boldsymbol{\gamma}, \boldsymbol{\beta}) = \boldsymbol{\gamma} \cdot (\boldsymbol{z}^{(l)} - \boldsymbol{\mu})/\sqrt{\boldsymbol{\sigma}} + \boldsymbol{\beta}$ for NORM-TUNE, AP can be interpreted as a variant of NORM-TUNE. **Fig.** 3 illustrates the connection. Specifically,

• *Conditions for CNNs*: when AP's perturbations are consistent across all feature map units, the unit-scaling BatchNorm-based NORM-TUNE closely mirrors the formulation of AP, differentiated merely by a linear mapping plus a bias. This equivalence becomes apparent when relating $\boldsymbol{W}^{(l)}\boldsymbol{\delta}^{(l)}$ to $\boldsymbol{\beta} - \boldsymbol{\gamma} \cdot \boldsymbol{\mu}/\sqrt{\boldsymbol{\sigma}}$, especially when $\boldsymbol{\gamma}/\sqrt{\boldsymbol{\sigma}} = 1$, supposing $\boldsymbol{W}^{(l)}$ as the weight for the $l$-th layer.

• *Conditions for ViTs*: assuming uniform perturbations across tokens and mean value is consistent across data dimensions within a batch, AP reduces to the unit-scaling LayerNorm-based NORM-TUNE, using the mean as the bias. This can be represented as $\boldsymbol{\delta}^{(l)} = \boldsymbol{\beta} - \boldsymbol{\mu}$, given $\boldsymbol{\gamma}/\sqrt{\boldsymbol{\sigma}} = 1$.

Due to more flexible perturbations of AP, such a connection exhibits increased power of AP than NORM-TUNE in terms of a larger parameter space for bias tuning. We theoretically summarize the above connection as Proposition 1 in Appx. C.2. Meanwhile, we remark that another key difference of AP compared to NORM-TUNE is that no parameters of the model backbones need to be altered during training. This differentiates "prompting" technology from any other PEFT methods by purely using the knowledge extracted by the pretrained model backbone alone. In the realm of PEFT, recent research has shown that LayerNorm-based NORM-TUNE serves as a robust baseline of model adaptation for ViTs (Basu et al., 2023). Beyond that, we will show that AP can surpass NORM-TUNE and remain effective for CNNs.

## 4 A DEEP DIVE INTO AP: LAYER AND ARCHITECTURE EFFECT

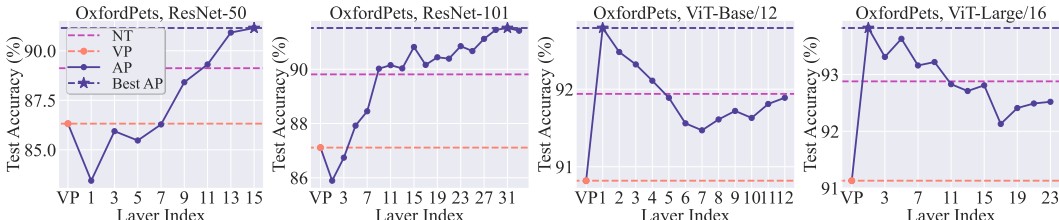

Figure 4: Layer preference of AP with different model architectures on OxfordPets (Parkhi et al., 2012). CNNs and ViTs exhibit opposite layer preferences. Results on more datasets are provided in Fig. A1.

### 4.1 INFLUENCE OF MODEL ARCHITECTURE ON AP'S LAYER PREFERENCE.

Our preliminary findings in Fig. 2 suggest that the effectiveness of AP may be contingent on the specific layer where it is installed. To acquire a deeper understanding of this characteristic and its association with model architecture, we examine two commonly-used model architectures: ResNet and ViT. **Fig. 4** follows and expands Fig. 2 by covering the additional models, namely ResNet-50, ViT-Base/12, and ViT-Large/16, and showcasing the transfer learning accuracy enabled by AP on the downstream dataset OxfordPets as a function of the layer index to which AP is applied. As we can see, a key observation is that *ResNets and ViTs exhibit contrasting layer preferences for* AP, where ★ indicates the best performance of AP in Fig. 4 under each architecture. Specifically, CNNs exhibit a preference for AP in their *deeper* layers, while ViTs tend to favor AP in their *shallower* layers. Moreover, it is worth noting that within the comfort layer zone, the performance of AP consistently outperforms NORM-TUNE. These insights underscore the significant potential of AP as an effective PEFT method, which will be further elucidated in Sec. 5.

**Dissecting CNNs and ViTs: AP prioritizes 'global' features over 'local' features.** To unpack the intriguing AP's layer preference behavior above, we next examine the features captured by different layers of CNNs and ViTs. To this end, we first employ the Centered Kernel Alignment (CKA)-based feature similarity analysis (Cortes et al., 2012) to measure the layer-wise representation similarity between CNNs and ViTs, *e.g.*, ResNet-101 and

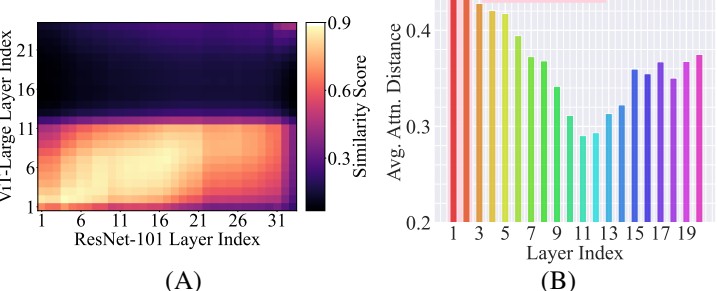

Figure 5: Features dissection to understand the layer effect of AP on OxfordPets dataset. (A) CKA-based feature similarity comparison between ViT-Large/16 and ResNet-101. (B) The average attention distance across all the heads of different layers of ViT-Large/16. A larger distance signifies a more globally-focused attention, indicative of global features.

ViT-Large/16 in **Fig. 5** (A). As we can see, the deep features of ResNet-101 predominantly align with the middle layers of ViT-Large/16. This concurs with the observations made in (Raghu et al., 2021), which suggest that ViTs have the capability to capture features reminiscent of the deeper layers of CNNs even within their relatively early layers. In addition, as indicated by network dissection analysis

for CNNs (Bau et al., 2017), it is known that CNNs tend to prioritize low-level visual concepts, *i.e.*, *local features* like color and texture, in their shallower layers. In contrast, they transition to high-level, class-discriminative concepts, encompassing *global features* like scenes and objects in deeper layers.

Drawing upon the analyses presented above and insights in Fig. 4, *we hypothesize* that AP exhibits a preference for deep layers in CNNs and shallow layers in ViTs, which can be attributed to the models' inclinations toward global features over local features. To bolster our hypothesis, we investigate how global information is distributed across the layers of ViTs. We employ a methodology used in (Raghu et al., 2021) and (Walmer et al., 2023) to compute the average attention distance between the position of query tokens and the locations they attend to with the query within each self-attention head in ViTs. This analysis unveils how each self-attention layer contributes to the balance between local and global information in the overall representation. In **Fig. 5** (B), we present the average attention distance across 16 attention heads for with different layer indices of a pretrained ViT-Large/16. A general trend can be observed: the distribution of the sorted attention distance moves firstly downwards (layer index from 1 to layer 12). This implies that the ratio of the global features captured by attention in general decreases. When the layer index is larger than 15, the global feature ratio slightly increases. This trend roughly aligns well with the patterns observed in Fig. 4. These observations underscore our claim that AP's layer preference is influenced by the presence of global features. We provide theoretical support in the following section to support the layer and architecture effect. In particular, we focus on the more challenging part of ViTs, since the study on CNNs is abundant.

## 4.2 WHY DOES LAYER AND ARCHITECTURE EFFECT HOLD IN THEORY?

From a perspective of generalization, we discuss the following layer and architecture effect for ViTs:

> *To achieve the desired generalization performance, shallow-layer* AP *tuning requires less sample complexity than deep-layer ones for ViTs.*

To show this, we present the theoretical setups that satisfy the conditions of global features for ViTs, followed by the generalization analysis with sample complexity bound in Theorem 1.

**Problem setup.** Following existing theoretical works (Li et al., 2023a; Oymak et al., 2023; Tarzanagh et al., 2023), we study a binary classification problem using a single-head two-layer ViT as the pretrained model with the dataset $\{x_n, y_n\}_{n=1}^N$. $y_n \in \{+1, -1\}$. Each data $x_n \in \mathbb{R}^{d \times P}$ consists of $P$ tokens. The training is implemented by a mini-batch stochastic gradient descent (SGD) with the loss $\ell(x_n, y_n)$. The generalization performance is evaluated by the population risk $\mathbb{E}[\ell(x_n, y_n)]$.

**Data assumption.** Each token of $x_n$ is formulated as a pattern added with a Gaussian noise following $\mathcal{N}(0, \sigma^2)$, $\sigma \leq O(1/P)$. We consider four patterns $\{v_1, v_2, v_3, v_4\}$ in total. In each $x_n$, only one token corresponds to either $v_1$ or $v_2$, named discriminative patterns that decide the label. Other $P-1$ tokens correspond to either $v_3$ or $v_4$, named non-discriminative patterns that work as the image background. For instance, if one token is the noisy version of $v_1$ ($v_2$), then $y^n = 1$ ($y^n = -1$).

**Pretrained model assumption.** The pretraining is assumed to learn a task where all four patterns are key features. Following the recent SOTA theoretical finding (Shi et al., 2022; Li et al., 2023a) that hidden neurons learn discriminative features, we set the MLP neurons as features that appear in the $l$-th layer and assign the value of the linear heads accordingly. To characterize the global features shown in **Fig. 5**, we assume the key and vector matrices to be scalings of permutation matrices. The details about the data and model assumptions can be found in Appx. C.3. Given a set of queries $q_1, \cdots, q_P$ and keys $k_1, \cdots, k_P$ for an attention head, We formally define the *average attention distance* mentioned in **Fig. 5** as $\sum_{i=1}^P |i - \arg\max_{j \in [P]} \langle k_j, q_i \rangle|/P$, i.e., the average distance between the query $q_i$ and the key $k_j$ that has the largest inner product with $q_i$, $i, j \in [P]$. Assuming the discriminative key and value are away from the discriminative query with a distance of $d_A \geq 1$, we have the following Lemma on decreasing the average attention distance.

**Lemma 1** *The average attention distance defined above decreases from $(1 + d_A)/P$ to $1/P$ after the 1st layer of the simplified two-layer ViT.*

Lemma 1 supports our empirical observation in **Fig. 5** (c) of decreasing attention distance values in deep layers in ViT. Such a decrease results in an increased sample complexity for guaranteed generalization, summarized in the following theorem.

**Theorem 1** *For the two-layer ViT, training with SGD returns a model with zero generalization error as long as the batch size $B \geq \Omega(1)$ and the required number of samples $N$ satisfy (1)*

$N \geq N_1 = \Theta(P)$ *if adding* AP *to the 1st layer; (2)* $N \geq N_2 = \Theta(P^2 \log P)$ *if adding* AP *to the 2nd layer.* $N_2$ *is order-wise larger than* $N_1$.

Theorem 1 shows deep-layer AP requires more training samples than the shallow-layer AP to achieve the same generalization, as shown by the dashed line in **Fig. 6**. Accordingly, with the same number of training samples and the setup, shallow-layer AP has better generalization than deep-layer AP. The proof of Theorem 1 can be found in Sec. C.4. The basic proof idea is that for AP in the shallow layer, a trained prompt with a norm of $\Theta(P)$ that removes non-discriminative patterns is enough to make all tokens attend to discriminative tokens. Thus, the amount of global features does not decrease. This can ensure zero generalization by abundant global features. For AP in deep layers, however, given Lemma. 1, a lack of global features leads to an evident mismatch between discriminative tokens in the 2nd-layer self-attention. Hence, a trained prompt with a norm of $\Theta(P^2 \log P)$ is necessary to direct the attention to focus on discriminative tokens. We then show the sample complexity bound is proportional to the magnitude of the trained prompt in these two cases.

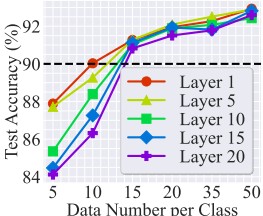

Figure 6: Sample complexity study of AP in different layers on OxfordPets with ViT-Large/16.

# 5 EXPERIMENTS

## 5.1 EXPERIMENT SETUP

**Datasets and models.** We utilize two commonly used architectures for the source datasets: ResNet-101 from the ResNet family (He et al., 2016) and ViT-Large/16 from the ViT family (Dosovitskiy et al., 2020). Both are pretrained on ImageNet-1K (Russakovsky et al., 2015). In the target domain, we consider over 20 datasets from transfer learning benchmarks FGVC (Maji et al., 2013) and VTAB (Zhai et al., 2019). In VTAB, we consider both *full-data* and *few-shot* (VTAB-1k) regimes. In addition, we also consider other commonly used datasets (Chen et al., 2023b) for transfer learning like CIFAR-10 (Krizhevsky et al., 2009), UCF101 (Soomro et al., 2012), GTSRB (Houben et al., 2013), Food101 (Bossard et al., 2014), and Waterbirds (Sagawa et al., 2019). More details on the datasets and the benchmarks can be found in Appx. A.

We cover three types of baselines in transfer learning. *First*, we primarily compare AP to the parameter-efficient learning methods designed for both CNNs and ViTs, including LINEAR-PROBE that only finetunes the classification head with a fixed feature extractor, the conventional (input-level) VP (Bahng et al., 2022a) and NORM-TUNE (Basu et al., 2023) that tunes *all* the normalization layers in a model. *Second*, we select FULL-FINETUNE as our reference method due to its superior accuracy; it fine-tunes the entire pretrained model, albeit being the most computationally expensive option. *Lastly*, we consider another 9 SOTA PEFT baselines used in ViT families: VPT (Jia et al., 2022), LoRA (Hu et al., 2021), ADAPTER (Chen et al., 2022b), BIAS (Zaken et al., 2021), NORM-TUNE (Basu et al., 2023), ATTNSCALE (Basu et al., 2023), and ADAPTERFORMER (Chen et al., 2022b). These methods help us to establish the ranking of AP in the PEFT domain.

**Implementation, training, and evaluations.** We implement AP at the input of the third-to-last ResNet block in ResNet-101 and the third Transformer block in ViT-Large/16, based on the layer effect in Fig. 4. During training, all the methods are trained for 100 epochs using the Cross-Entropy loss with an Adam optimizer (Kingma & Ba, 2015). Hyperparameters, including learning rates, are determined through a search process for each method, and the implementation details are summarized in Appx. A. During evaluation, we compare different methods in terms of their performance (testing accuracy) and efficiency. In particular, we depict the **efficiency portrait** of a method from the following 4 different perspectives: (1) tunable parameter number, (2) memory cost, (3) train time per epoch, and (4) throughput for inference efficiency; see Tab. 2 for more details.

## 5.2 EXPERIMENT RESULTS

**AP is not only effective but also efficient.** We examine the performance of the proposed AP in the full-data regime below. *Two key observations* can be drawn from experiment results: (1) AP consistently outperforms baselines across the majority of datasets from various dataset benchmarks, in particular with a significant improvement over VP (see Tab. 1); (2) AP demonstrates remarkable efficiency across various efficiency metrics, establishing itself as a cost-effective and potent method, as highlighted in Tab. 2.

Table 1: Performance comparison of various methods on 19 datasets from different benchmarks. Three parameter-efficient baselines (denoted by ○) are compared to AP due to their high relevance, where the best performance is highlighted in **bold**. The most computationally intensive FULL-FINETUNE (denoted by ●) serves as the performance reference. Each accuracy value is averaged over 5 independent trials, with the variance omitted due to its negligible values ($\leq 0.3\%$). The "Average" column represents the averaged accuracy of each method over all the datasets in each row.

| Architecture | Benchmark | CUB200 | StanfordDog | StanfordCars | NA-Birds | OxfordFlowers | CIFAR-100 | Caltech-101 | DTD | Flowers102 | OxfordPets | SVHN | SUN397 | Camelyon | EuroSAT | CIFAR-10 | GTSRB | UCF101 | Food101 | Waterbirds | Average |
|---|---|---|---|---|---|---|---|---|---|---|---|---|---|---|---|---|---|---|---|---|---|
| ResNet-101 | ● FULL-FINETUNE | 88.91 | 90.13 | 87.76 | 84.45 | 99.98 | 92.24 | 99.13 | 79.97 | 99.81 | 90.49 | 97.14 | 79.19 | 91.13 | 99.13 | 97.24 | 97.68 | 88.32 | 82.72 | 96.69 | 91.69 |
| | ○ LINEAR-PROBE | 63.76 | 86.63 | 49.62 | 52.09 | 82.01 | 73.87 | 90.58 | 61.35 | 93.14 | 91.17 | 66.30 | 54.51 | 83.36 | 95.84 | 92.25 | 79.64 | 71.03 | 64.31 | 88.11 | 75.76 |
| | ○ NORM-TUNE | 66.39 | 87.59 | **67.64** | 56.72 | 66.50 | **82.58** | 91.32 | 63.53 | 92.85 | 89.81 | **95.26** | 54.56 | 84.42 | 96.14 | 93.90 | **96.43** | 69.44 | **72.54** | **88.95** | 79.81 |
| | ○ VP | 65.72 | 86.91 | 51.04 | 54.23 | 78.50 | 72.01 | 93.51 | 63.12 | 90.17 | 87.93 | 80.68 | 54.97 | 83.71 | 95.44 | 92.55 | 83.18 | 66.30 | 57.89 | 86.71 | 76.03 |
| | ○ AP (ours) | **69.42** | **87.79** | 59.06 | **58.31** | **85.14** | 76.94 | **94.85** | **69.80** | **95.13** | **91.31** | 87.30 | **56.83** | **84.91** | **97.21** | **94.08** | 90.43 | **73.96** | 68.12 | 88.13 | **80.45** |
| ViT-Large/16 | ● FULL-FINETUNE | 89.79 | 93.31 | 89.42 | 84.75 | 99.91 | 93.19 | 99.25 | 75.30 | 99.39 | 93.35 | 98.13 | 79.31 | 91.93 | 97.92 | 98.30 | 97.90 | 89.25 | 86.16 | 97.93 | 92.34 |
| | ○ LINEAR-PROBE | 84.69 | 86.11 | 65.24 | 75.71 | 99.40 | 88.55 | 97.01 | 73.31 | 99.24 | 91.15 | 65.79 | 72.37 | 84.05 | 97.26 | 98.13 | 80.72 | 83.02 | 83.02 | 94.16 | 85.20 |
| | ○ NORM-TUNE | 85.90 | 89.76 | **75.61** | 78.78 | 99.35 | 90.69 | 98.01 | 78.90 | 99.76 | 92.88 | 88.30 | 73.57 | 79.82 | 97.17 | 98.44 | 90.86 | 85.15 | 83.21 | 94.36 | 88.45 |
| | ○ VP | 85.24 | 87.02 | 67.64 | 76.20 | 99.32 | 89.44 | 97.81 | 77.72 | 99.72 | 91.31 | 85.70 | 74.33 | 84.27 | 97.85 | **98.80** | 89.09 | 84.67 | 82.23 | **95.03** | 87.54 |
| | ○ AP (ours) | **86.74** | **90.83** | 69.41 | **79.83** | 99.70 | **90.96** | **98.99** | 78.96 | **99.84** | **93.89** | **88.87** | **75.44** | **86.99** | **98.33** | 98.54 | **91.49** | **86.80** | 84.04 | 94.60 | **89.17** |

**Tab. 1** shows the performance of AP vs. the baselines VP, NORM-TUNE, LINEAR-PROBE, and FULL-FINETUNE. As we can see, AP consistently outperforms VP in *all* the 19 datasets. Notably, AP yields an increase in the average accuracy of over $4\%$ and $1.5\%$ compared to VP for both ResNet-101 and ViT-Large/16. In some datasets, such as StanfordCars, SVHN and GT-SRB using ResNet-101, this advantage can increase to $7\%\sim9\%$. AP also remains effective compared to NORM-TUNE, which has proven to be a strong PEFT method for ViT families in (Basu et al., 2023). AP performs the best in 13 and 15 out of 19 datasets for ResNet-101 and ViT-Large/16, respectively. Although FULL-FINETUNE remains the best-performing in most

Table 2: An overview of the methods considered in this work. The efficiency analysis is based on the model-data setting (ViT-Large, CIFAR-10) with a batch size of 128, and time consumption is evaluated using a single RTX-A6000 GPU. For each metric, we use ↑ or ↓ to indicate whether a larger smaller value is favored for each metric.

| Method | Param. Efficiency Parameter # (M) ↓ | Train-Time Efficiency Memory Cost (G) ↓ | Time Cost (s/epoch) ↓ | Troughput (image/s) ↑ |
|---|---|---|---|---|
| **ResNet-101** | | | | |
| FULL-FINETUNE | 44.5 | 10.32 | 118 | 41.47 |
| LINEAR-PROBE | 0.02 | 6.2 | 39 | 41.33 |
| NORM-TUNE | 0.13 | 11.7 | 83 | **41.45** |
| VP | **0.12** | 12.2 | 72 | 40.59 |
| AP | **0.12** | 6.3 | 41 | 41.36 |
| **ViT-Large/16** | | | | |
| FULL-FINETUNE | 304.33 | 41.5 | 520 | 79.58 |
| LINEAR-PROBE | 0.01 | 9.7 | 121 | 79.64 |
| NORM-TUNE | **0.06** | 29.5 | 285 | **79.51** |
| VP | 0.11 | 35.9 | 280 | 77.14 |
| AP | 0.16 | 31.6 | 262 | 79.48 |

datasets, AP still manages to surpass FULL-FINETUNE in some datasets, such as OxfordPets for ResNet-101, and DTD for ViT-Large/16. Most importantly, AP is much more efficient than FULL-FINETUNE, as illustrated below.

**Tab. 2** demonstrates the efficiency profile of different methods under different metrics. Two key insights can be drawn from the results. *First*, in comparison to VP, AP demonstrates superior efficiency in terms of memory (reduced memory overhead), time (decreased training duration), and inference (increased throughput) for both ResNet-101 and ViT-Large/16. This superiority is maintained while operating at a comparable parameter efficiency, marked by a negligible tunable ratio difference of less than $0.05\%$. This trend is amplified for ResNet-101, as evidenced by the significant reductions in memory usage (6.3 G for AP vs. 12.2 G for VP) and training duration (41 s/epoch for AP vs. 72 s/epoch for VP). This efficiency arises from the AP's preference towards deeper layers over shallower ones in ResNet-101, resulting in reduced back-propagation overhead for most of the network. *Second*, when compared to NORM-TUNE, although AP consumes slightly higher memory cost for ViT-Large/16, it achieves higher training efficiency for ResNet-101 and ViT-Large/16. This is due to that, while NORM-TUNE possesses a small tunable parameter ratio, these parameters are dispersed throughout the network, leading to a more expensive back-propagation process. Although no significant difference is observed in throughput, we will show later in Tab. 4 that AP enjoys high throughput efficiency compared to other PEFT methods.

**How does the downstream dataset scale affect AP?** To study the effect brought by the downstream data scales, we follow the setting of (Jia et al., 2022) and examine the performance of different methods under the few-shot setting on VTAB-$1K$. In particular, for each of the 19 datasets in the VTAB benchmark, only 1000 data samples are available for training. **Tab. 3** shows that AP makes a

Table 3: Performance comparison of various methods in the few-shot setting on the VTAB-$1K$ benchmark. Other settings follow Tab. 1.

| Architecture | Benchmark | VTAB-Natural | | | | | | | VTAB-Specialized | | | | VTAB-Structured | | | | | | | | |
|---|---|---|---|---|---|---|---|---|---|---|---|---|---|---|---|---|---|---|---|---|---|
| | | Caltech101 | CIFAR-100 | DTD | Flowers102 | OxfordPets | Sun397 | SVHN | Camelyon | EuroSAT | Resisc45 | Retinopathy | Clevr-Count | Clevr-Dist | DMLab | dSpr-Loc | dSpr-Ori | KITTI-Dist | sNORB-Azim | sNORB-Elev | Average |
| ResNet-101 | ● FULL-FINETUNE | 89.99 | 45.17 | 63.78 | 84.29 | 89.82 | 41.09 | 67.79 | 84.92 | 74.57 | 91.37 | 74.14 | 58.11 | 60.99 | 43.61 | 67.05 | 40.45 | 78.34 | 33.64 | 36.38 | 64.50 |
| | ○ LINEAR-PROBE | 83.87 | 39.13 | 53.09 | 70.89 | 85.15 | 28.14 | 43.44 | 78.65 | 69.43 | 90.78 | 69.31 | 35.91 | 36.48 | 35.75 | 34.76 | 19.51 | 65.68 | 16.91 | 23.39 | 51.12 |
| | ○ NORM-TUNE | 85.61 | 35.78 | 47.71 | 56.64 | 78.10 | 10.10 | **68.67** | **83.16** | 61.10 | 90.50 | **72.44** | 37.54 | 55.24 | **40.04** | **60.89** | 20.33 | 65.54 | 24.86 | 25.96 | 53.70 |
| | ○ VP | 84.73 | **43.01** | 57.55 | 76.91 | 87.03 | 28.75 | 55.47 | 75.15 | 70.27 | 89.26 | 69.08 | 36.70 | 54.24 | 34.48 | 42.41 | 20.32 | 63.71 | 17.93 | 26.93 | 54.42 |
| | ○ AP | **87.49** | 39.80 | **63.62** | **81.44** | **88.74** | **34.83** | 65.92 | 78.91 | **74.19** | **91.44** | 71.18 | **40.20** | **55.26** | 38.95 | 54.68 | **21.98** | **72.86** | **26.24** | **28.77** | **58.76** |
| ViT-Large/16 | ● FULL-FINETUNE | 93.34 | 76.03 | 75.74 | 99.88 | 93.72 | 59.06 | 68.70 | 86.70 | 82.84 | 93.54 | 82.22 | 55.42 | 60.33 | 48.23 | 83.62 | 52.77 | 78.06 | 30.40 | 29.95 | 71.08 |
| | ○ LINEAR-PROBE | 89.37 | 62.98 | 70.02 | 93.42 | 91.22 | 53.68 | 45.28 | 80.52 | 80.34 | 91.64 | 70.43 | 38.15 | 35.26 | 40.74 | 21.84 | 29.42 | 62.54 | 14.59 | 23.09 | 57.60 |
| | ○ NORM-TUNE | 91.10 | **65.20** | 72.36 | 98.64 | 91.38 | 55.14 | 47.21 | **82.50** | 82.34 | **93.94** | 71.74 | 42.83 | 44.59 | **41.21** | 35.64 | **32.08** | 63.43 | 16.52 | 24.12 | 60.68 |
| | ○ VP | 90.06 | 63.16 | 71.59 | 95.35 | 91.20 | 54.45 | 46.26 | 81.82 | 81.45 | 92.25 | 71.03 | 41.03 | **45.49** | 39.94 | 32.52 | 30.29 | 62.68 | 15.59 | 23.13 | 59.96 |
| | ○ AP | **91.40** | 64.40 | **72.61** | **99.50** | **91.46** | **56.67** | **49.43** | 81.41 | **82.76** | 93.14 | **71.99** | **43.26** | 38.09 | 40.57 | **42.44** | 31.83 | **65.40** | **18.29** | **25.96** | **61.06** |

distinguishable improvement over the baselines VP and NORM-TUNE in the few-shot setting. As we can see, AP achieves a performance boost of over $1\%$ than VP using ViT-Large/16 and this advantage increases to $4.3\%$ in the case of ResNet-101. This demonstrates that directly steering the intermediate features can be more effective when facing data scarcity.

**Comparing AP with more SOTA PEFT baselines.** To demonstrate the applicability of AP as an effective and generic PEFT method, we compare AP with more SOTA PEFT methods in **Tab. 4**. As we can see, even when compared to the strongest PEFT baselines, AP still remains a competitive method in terms of both accuracy and efficiency. For example, AP ranks roughly 2∼4 in terms of accuracy among the 8 PEFT methods considered in this work (including VP and NORM-TUNE). More importantly, AP ranks the first in terms of all

Table 4: Performance comparison between AP and more SOTA PEFT methods on ViT-Large/16. Experiment settings follow Tab. 1, and Tab. 2.

| | Accuracy | | | Efficiency | | | |
|---|---|---|---|---|---|---|---|
| | Full-Data | | | Train-Time Efficiency | | | |
| | FGVC | VTAB | Others | Param. # | Memory | Time | Throughput |
| Number of tasks | 5 | 9 | 5 | - | - | - | - |
| FULL-FINETUNE | 91.43 | 91.97 | 93.91 | 304.33 | 41.5 | 520 | 79.58 |
| LINEAR-PROBE | 82.23 | 78.90 | 87.81 | 0.01 | 9.7 | 121 | 79.64 |
| BIAS | 85.32 | 89.84 | 90.41 | 0.29 | 32.9 | 297 | **79.43** |
| LoRA | 86.87 | 89.81 | 91.45 | 1.00 | 33.1 | 363 | **79.43** |
| VPT | 86.34 | 89.24 | 90.14 | 0.25 | 33.7 | 334 | 76.35 |
| ADAPTER | 87.06 | 89.44 | 91.21 | 2.17 | 32.4 | 357 | 63.39 |
| ADAPTERFORMER | **89.18** | **90.69** | **92.08** | 0.65 | 32.3 | 289 | 23.69 |
| AP | 85.30 | 90.25 | 91.09 | **0.16** | **31.6** | **262** | **79.43** |

of the four perspectives of efficiency. In contrast, the best performance of the ADAPTERFORMER comes at a cost of three times lower throughput efficiency. This is due to that extra modules introduce significantly more computations during the inference.

**Other ablation studies.** We provide additional experiment results in Appx. B. In particular, we showcase the effectiveness of AP across a broader spectrum of model architectures and its compatibility with models pretrained using various pretraining methods. We also provide layer effect analysis similar to Fig. 4 on more datasets.

**Discussion and limitations.** It is worth noting that a potential limitation of AP lies in its implicit reliance on the size of the pretrained model as a factor for achieving superior accuracy. Specifically, for compact models like ResNet-18 and ViT-Tiny, while AP enhances the performance of VP, it does not outperform NORM-TUNE. This observation suggests that AP may primarily utilize downstream data to guide or "direct" the existing learned knowledge obtained during pretraining, rather than actively acquiring new knowledge. However, we believe that this limitation does not prevent AP from future applications to larger foundational vision models.

# 6 CONCLUSION

In this paper, we delve into AP (activation prompt) as a means to enhance VP. We unveil that extending VP to AP yields improved empirical performance and establishes a connection with normalization tuning NORM-TUNE. Additionally, we investigate the layer preference of AP on CNNs and ViTs both empirically and theoretically. Our experimentation across 29 datasets clearly illustrates the efficiency of AP compared to other PEFT methods and its superiority over VP.

## REPRODUCIBILITY STATEMENT

The authors have made an extensive effort to ensure the reproducibility of algorithms and results presented in the paper. *First*, the details of the experiment settings have been elaborated in Sec. 5 and Appx. A. In this paper, 29 datasets are studied and the details about each dataset is described in Tab. A1. The evaluation metrics are also clearly introduced in Sec. 5. *Second*, 9 PEFT methods are studied in this work. The implementation details of all the baseline methods are clearly presented in Appx. A. For our proposed AP, we include the implementation details in Sec. 5 and Appx. A. *Third*, all the results are based on 5 independent trials with different random seeds to guarantee the reliability of the results. *Fourth*, codes are included in the supplementary material.

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

# APPENDIX

## A EXPERIMENT SETTING DETAILS

**Datasets.** We consider 29 downstream image classification tasks in the target domain across various domains. We show each dataset's attributes in Tab. A1.

| Dataset | Train Size | Test Size | Class Number | Batch Size | Reference |
|---|---|---|---|---|---|
| **Full-Data Setting** | | | | | |
| Flowers102 | 4093 | 2463 | 102 | 128 | (Nilsback & Zisserman, 2008) |
| DTD | 2820 | 1692 | 47 | 128 | (Cimpoi et al., 2014) |
| UCF101 | 7639 | 3783 | 101 | 128 | (Soomro et al., 2012) |
| Food101 | 50500 | 30300 | 101 | 128 | (Bossard et al., 2014) |
| SVHN | 73257 | 26032 | 10 | 128 | (Netzer et al., 2011) |
| GTSRB | 39209 | 12630 | 43 | 128 | (Houben et al., 2013) |
| EuroSAT | 13500 | 8100 | 10 | 128 | (Helber et al., 2019) |
| OxfordPets | 2944 | 3669 | 37 | 128 | (Parkhi et al., 2012) |
| StanfordCars | 6509 | 8041 | 196 | 128 | (Krause et al., 2013) |
| SUN397 | 15888 | 19850 | 397 | 128 | (Xiao et al., 2010) |
| CIFAR10 | 50000 | 10000 | 10 | 128 | (Krizhevsky et al., 2009) |
| CIFAR100 | 50000 | 10000 | 100 | 128 | (Krizhevsky et al., 2009) |
| CUB-200-2011 | 5394 | 5794 | 200 | 128 | (Wah et al., 2011) |
| NA-Birds | 21536 | 24633 | 55 | 128 | (Van Horn et al., 2015) |
| StanfordDog | 10800 | 8580 | 120 | 128 | (Khosla et al., 2011) |
| OxfordFlowers | 1020 | 6149 | 102 | 128 | (Nilsback & Zisserman, 2008) |
| Waterbirds | 4795 | 5794 | 2 | 128 | (Sagawa et al., 2019) |
| Caltech101 | 4128 | 2465 | 102 | 128 | (Li et al., 2006) |
| Camelyon | 262144 | 32768 | 2 | 128 | (Veeling et al., 2018) |
| **Few-Shot Setting (VTab-1k)** | | | | | |
| CIFAR-100 | 1000 | 10000 | 100 | 128 | (Krizhevsky et al., 2009) |
| Caltech101 | 1000 | 6084 | 102 | 128 | (Li et al., 2006) |
| DTD | 1000 | 47 | 1880 | 128 | (Cimpoi et al., 2014) |
| Flowers102 | 1000 | 6149 | 102 | 128 | (Nilsback & Zisserman, 2008) |
| OxfordPets | 1000 | 3669 | 37 | 128 | (Parkhi et al., 2012) |
| SVHN | 1000 | 26032 | 10 | 128 | (Netzer et al., 2011) |
| Sun397 | 1000 | 21750 | 397 | 128 | (Xiao et al., 2010) |
| Patch Camelyon | 1000 | 32768 | 2 | 128 | (Veeling et al., 2018) |
| EuroSAT | 1000 | 5400 | 10 | 128 | (Helber et al., 2019) |
| Resisc45 | 1000 | 6300 | 45 | 128 | (Cheng et al., 2017) |
| Retinopathy | 1000 | 42670 | 5 | 128 | (Kaggle & EyePacs, 2015) |
| Clevr/count | 1000 | 15000 | 8 | 128 | (Johnson et al., 2017) |
| Clevr/distance | 1000 | 15000 | 6 | 128 | (Johnson et al., 2017) |
| DMLab | 1000 | 22735 | 6 | 128 | (Beattie et al., 2016) |
| KITTI/distance | 1000 | 711 | 4 | 128 | (Geiger et al., 2013) |
| dSprites/location | 1000 | 73728 | 16 | 128 | (Matthey et al., 2017) |
| dSprites/orientation | 1000 | 73728 | 16 | 128 | (Matthey et al., 2017) |
| SmallNORB/azimuth | 1000 | 12150 | 18 | 128 | (LeCun et al., 2004) |
| SmallNORB/elevation | 1000 | 12150 | 9 | 128 | (LeCun et al., 2004) |

Table A1: Dataset attributes and training configs through 29 target image-classification datasets.

**Implementation details.** As we stated in the main manuscript, we, by default, install AP to the input of the thrid-to-last ResNet block and the third Transformer block in ViT-Large/16. For LoRA (Hu et al., 2021), we use the rank $r = 10$ by default. For VPT (Jia et al., 2022), we use a prompt length of 10. We train all the methods for 1000 epochs using an Adam optimizer. For AP, we adopt a learning rate of 0.001 for ResNet family and 0.01 for ViT family without weight decay. For baselines, we adopt the learning rate suggested in the papers or official code repositories.

## B ADDITIONAL EXPERIMENT RESULTS

**Layer effect study on more datasets.** In Fig. A1, we demonstrate that the layer effects of AP demonstrated in Sec. 4 is general and apply to multiple datasets.

**Applying AP to various model architectures: A case study on CLIP.** In the main manuscript, we show that AP can effectively adapt large supervised pretrained models to downstream tasks and outperforms VP. In this section, we shift our focus from the vision source model to the vision-language model, specific to CLIP (contrastive language–image pretraining), which has received increasing attention in the area of VP Bahng et al. (2022b). Our experiments demonstrate that the proposed idea of AP works well even on steering a pretrained CLIP model without changing its

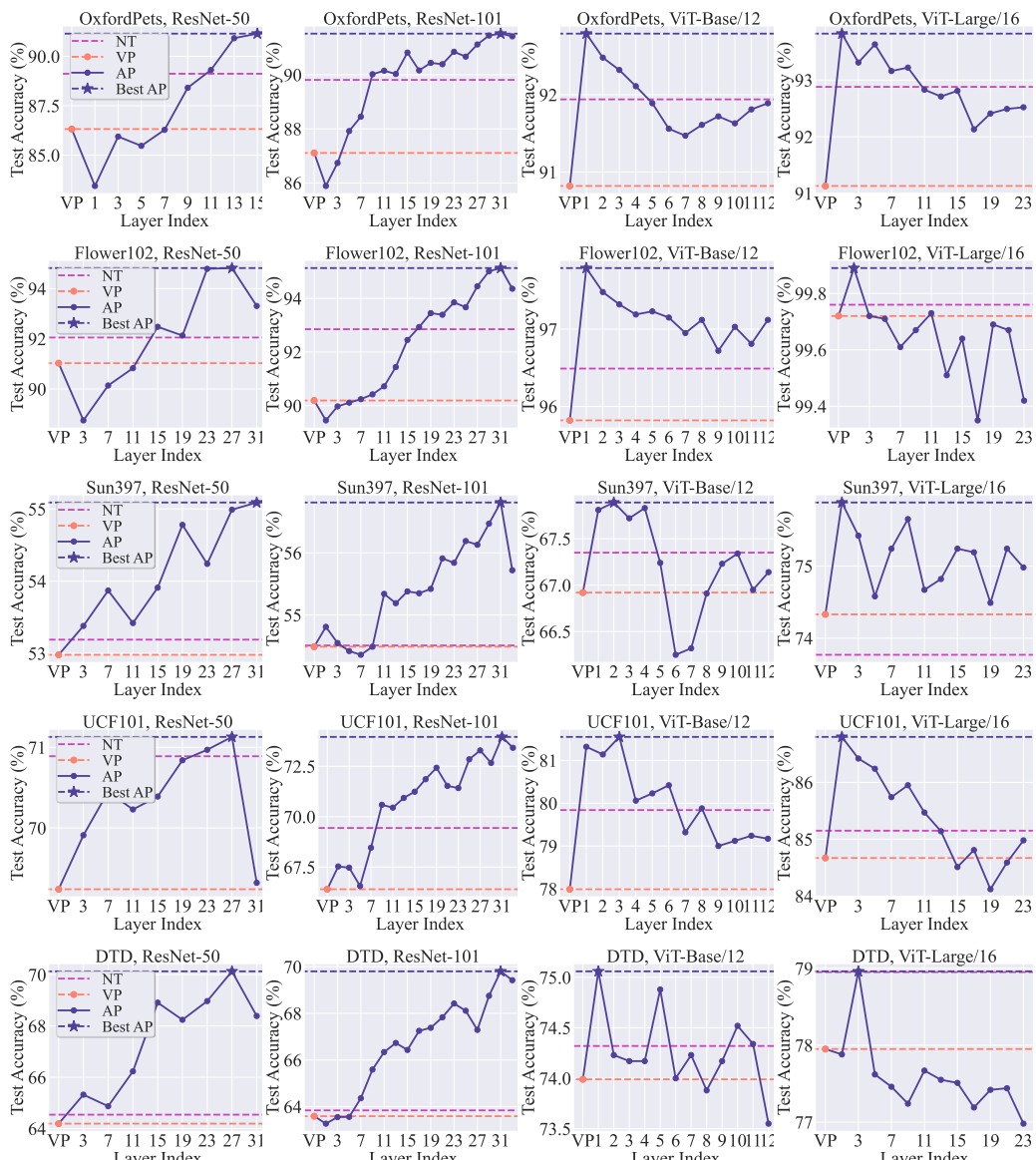

Figure A1: Layer preference of AP with different model architectures on different datasets. CNNs and ViTs exhibit opposite layer preferences.

parameters. In Fig. A2 and Tab. A2, we demonstrate that the main conclusions from Sec. 4 and Sec. 5 about AP still holds well on various datasets. Specifically, in Fig. A2, we show that the layer effect of AP still exists on the CLIP model. As the CLIP model uses a $\overline{\text{ViT as}}$ its backbone, the observed layer effect mimics that of a ViT-Large/16 as observed in Sec. 4. Specifically, AP prefers to be installed on shallow layers to deep ones in order to obtain the best performance. In Tab. A2, we demonstrate that in various datasets, AP can significantly outperform VP by $1\% \sim 6\%$. These experiments demonstrate the applicability of AP on various model types.

**Performance of AP in the precise experiment setting of VPT.** We conduct an ablation study to strictly follow the experiment settings of VPT, with these results included in Tab. A3. The performance of VPT is directly sourced from Tab. 1 of (Jia et al., 2022). As we can see, the performance as well as efficiency of AP positions itself between VPT-Shallow and VPT-Deep, with an average of 3% performance gain over VPT-Shallow and an average of 3.5% drop compared to VPT-Deep. Regarding these results, we would like to mention that the results of VPT reported in

Table A2: Performance comparison of VP and the proposed AP on CLIP and Swin-Transformer model with different dataset. The CLIP uses a ViT-B/32 as a backbone and we adopt Swin-B (with 12 Swin-Transformer blocks) pretrained on ImageNet. Other settings follows Tab. 1.

| Dataset | OxfordPets | DTD | EuroSAT | Flowers102 | UCF101 | Food101 | Waterbirds |
|---|---|---|---|---|---|---|---|
| | | | CLIP | | | | |
| VP | 81.97 | 64.43 | 95.54 | 83.74 | 70.42 | 79.61 | 72.42 |
| AP (Ours) | 83.82 | 69.42 | 96.43 | 85.52 | 76.42 | 82.43 | 79.32 |
| | | | Swin-Transformer | | | | |
| VP | 80.42 | 65.39 | 97.23 | 84.48 | 74.41 | 75.72 | 75.22 |
| AP (Ours) | 82.29 | 69.13 | 96.45 | 84.98 | 75.92 | 81.38 | 78.99 |

(a) CLIP  (b) Swin-Transformer

Figure A2: The layer effect of AP applied to a (a) CLIP model and (b) Swin-Transformer on the OxfordPets dataset.

Table 1 of (Jia et al., 2022) are selected based on its best prompt length per dataset, while AP sticks to the same hyper-parameters across all the datasets.

Table A3: Performance comparison of AP with other methods in the setting of VPT (Jia et al., 2022). Specifically, ViT-B/16 pretrained on supervised ImageNet-21k is adopted as the pretrained model. The numbers except AP are directly sourced from VPT (Jia et al., 2022).

| ViT-B/16 (85.8M) | Total Params | FGCV | VTAB-1k | | |
|---|---|---|---|---|---|
| | | | Natural | Specialized | Structured |
| FULL-FINETUNE | 24.02× | 88.54 | 75.88 | 83.36 | 47.64 |
| LINEAR-PROBE | 1.02× | 79.32 | 68.93 | 77.16 | 26.84 |
| VPT-SHALLOW | 1.04× | 84.62 | 76.81 | 74.66 | 46.98 |
| VPT-DEEP | 1.18× | 89.11 | 78.48 | 82.43 | 54.98 |
| AP (Ours) | 1.11× | 87.33 | 76.59 | 79.32 | 49.98 |

**Performance comparison with re-initialized classification head.** We carried out an ablation experiment using re-initialized classification head. This will influence the tunable parameter counts of LINEAR-PROBE and other methods involved. As we can see, the results in Tab. A4 are nearly identical to our previous findings in Tab. 4 that AP shows a competitive performance and efficiency compared with other strong PEFT baselines.

**Comparison to VPT with other prompt lengths.** We conducted an experiment to implement VPT-Deep using a smaller prompt token length 10 (VPT-10). The results, presented in Tab. A5, indicate that VPT-10's performance is comparable to VPT-50 in Tab. 4, albeit with enhanced efficiency.

**Layerwise comparison between AP and VPT-Deep.** We conduct an experiment for a more detailed layer-wise evaluation in Fig. A3. These additional results highlight a consistent layer-architecture influence on VPT-Deep, akin to what we initially observed in our original AP design. This outcome is not unexpected, considering that the implementation of VPT-Deep essentially converges with that

Table A4: Performance comparison between AP and SOTA PEFT methods on ViT-Large/16 with re-initialized classification head. Experiment settings follow Tab. 1, and Tab. 2.

| | Accuracy | | | Efficiency | | | |
|---|---|---|---|---|---|---|---|
| | **Full-Data** | | | **Train-Time Efficiency** | | | |
| | FGVC | VTAB | Others | Param. # | Memory | Time | Throughput |
| Number of tasks | 5 | 9 | 5 | - | - | - | - |
| FULL-FINETUNE | 91.43 | 91.97 | 93.91 | 304.33 | 41.5 | 520 | 79.58 |
| LINEAR-PROBE | 82.31 | 78.43 | 87.71 | 0.01 | 8.1 | 121 | 79.69 |
| BIAS | 85.49 | 89.47 | 90.85 | 0.29 | **27.4** | 297 | **79.51** |
| LORA | 86.49 | 89.74 | 91.49 | 1.00 | 32.5 | 363 | 71.47 |
| VPT | 86.15 | 90.13 | 90.88 | 1.24 | 37.2 | 397 | 72.91 |
| ADAPTER | 87.14 | 89.12 | 91.01 | 2.07 | 31.1 | 357 | 63.78 |
| ADAPTERFORMER | **89.24** | **90.49** | **92.21** | 0.65 | 31.1 | 289 | 23.82 |
| AP | 85.32 | 90.12 | 91.11 | **0.16** | 30.2 | **262** | 79.54 |

Table A5: Performance comparison between AP and VPT with different prompt lengths on ViT-Large/16. Experiment settings follow Tab. 1, and Tab. 4.

| | Accuracy | | | Efficiency | | | |
|---|---|---|---|---|---|---|---|
| | **Full-Data** | | | **Train-Time Efficiency** | | | |
| | FGVC | VTAB | Others | Param. # | Memory | Time | Throughput |
| Number of tasks | 5 | 9 | 5 | - | - | - | - |
| FULL-FINETUNE | 91.43 | 91.97 | 93.91 | 304.33 | 41.5 | 520 | 79.58 |
| LINEAR-PROBE | 82.23 | 78.90 | 87.81 | 0.01 | 9.7 | 121 | 79.64 |
| VPT-10 | 86.34 | 89.24 | 90.14 | 0.25 | 33.7 | 334 | 76.35 |
| VPT-50 | 86.05 | 89.97 | 90.64 | 1.24 | 38.6 | 397 | 72.84 |
| AP | 85.30 | 90.25 | 91.09 | **0.16** | **31.6** | **262** | 79.43 |

of AP when a specific network layer is selected for prompting. The key divergence lies in the prompt design approach: VPT-Deep favors concatenation, whereas AP opts for addition in prompt design. It is worth noting that, in the context of single-layer prompting, the efficacy of concatenation in prompt design is comparatively lower than that of addition.

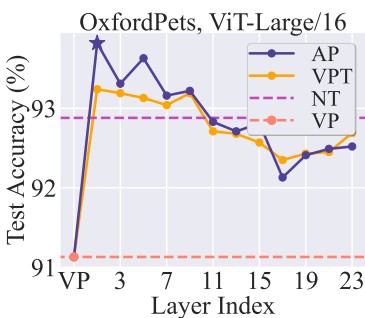

Figure A3: Layer-wise performance comparison between AP and VPT on OxfordPets.

**Ablation study on additional prompt types in AP.** We conduct additional experiments, with the findings presented in Tab. A6. We observed that the originally proposed AP outperforms its new prompt variants studied in Tab. A6 (AP-Product and AP-Concate). We speculate that the advantage of the originally proposed AP may stem from its intrinsic connection to NORM-TUNE, as discussed in the concluding part of Sec. 3.

**Comparison with additional PEFT methods.** We conduct an experiment and report the results of SSF in Tab. A7. In particular, we can see SSF is also a competitive method among all the baselines but is still under AdapterFormer. Compared to AP, SSF yields better performance for the FGVC benchmark but leads to slightly worse accuracy for the VTAB benchmark. In general, SSF ranks approximately the second or the third place among all the PEFT methods.

Table A6: Ablation study on AP with more prompt types. Specifically, instead of using additive prompt in the intermediate layer, AP-PRODUCT uses feature-wise product and AP-CONCATE adopts concatenating prompt.

| | Accuracy | | | Efficiency | | | |
| --- | --- | --- | --- | --- | --- | --- | --- |
| | **Full-Data** | | | | **Train-Time Efficiency** | | |
| | FGVC | VTAB | Others | Param. # | Memory | Time | Throughput |
| Number of tasks | 5 | 9 | 5 | - | - | - | - |
| FULL-FINETUNE | 91.43 | 91.97 | 93.91 | 304.33 | 41.5 | 520 | 79.58 |
| LINEAR-PROBE | 82.23 | 78.90 | 87.81 | 0.01 | 9.7 | 121 | 79.64 |
| BIAS | 85.32 | 89.84 | 90.41 | 0.29 | 32.9 | 297 | **79.48** |
| LoRA | 86.87 | 89.81 | 91.45 | 1.00 | 33.1 | 363 | **79.43** |
| VPT | 86.05 | 89.97 | 90.64 | 1.24 | 38.6 | 397 | 72.84 |
| ADAPTER | 87.06 | 89.44 | 91.21 | 2.07 | 32.4 | 357 | 63.39 |
| ADAPTERFORMER | **89.18** | **90.69** | **92.08** | 0.65 | 32.3 | 289 | 23.69 |
| AP-PRODUCT | 84.20 | 85.36 | 90.15 | **0.16** | **31.6** | **262** | **79.43** |
| AP-CONCATE | 83.29 | 82.42 | 89.13 | **0.12** | **31.4** | **261** | **79.47** |
| AP | 85.30 | 90.25 | 91.09 | **0.16** | **31.6** | **262** | **79.43** |

Table A7: Performance comparison of AP with more PEFT methods (SSF (Lian et al., 2022)). Experiment settings follow Tab. 1 and Tab. 4.

| | Accuracy | | | Efficiency | | | |
| --- | --- | --- | --- | --- | --- | --- | --- |
| | **Full-Data** | | | | **Train-Time Efficiency** | | |
| | FGVC | VTAB | Others | Param. # | Memory | Time | Throughput |
| Number of tasks | 5 | 9 | 5 | - | - | - | - |
| FULL-FINETUNE | 91.43 | 91.97 | 93.91 | 304.33 | 41.5 | 520 | 79.58 |
| LINEAR-PROBE | 82.23 | 78.90 | 87.81 | 0.01 | 9.7 | 121 | 79.64 |
| BIAS | 85.32 | 89.84 | 90.41 | 0.29 | 32.9 | 297 | **79.48** |
| LoRA | 86.87 | 89.81 | 91.45 | 1.00 | 33.1 | 363 | **79.43** |
| VPT | 86.05 | 89.97 | 90.64 | 1.24 | 38.6 | 397 | 72.84 |
| ADAPTER | 87.06 | 89.44 | 91.21 | 2.17 | 32.4 | 357 | 63.39 |
| ADAPTERFORMER | **89.18** | **90.69** | **92.08** | 0.65 | 32.3 | 289 | 23.69 |
| SSF | 87.32 | 89.43 | 92.21 | 0.48 | 34.7 | 299 | **79.49** |
| AP | 85.30 | 90.25 | 91.09 | **0.16** | **31.6** | **262** | **79.43** |

**Comparison with LoRA of different rank values.** We conduct additional experiments on the hyper-parameters of LoRA, namely the rank $r$. In Tab. 4, the rank $r$ is adopted to 10 by default. In Tab. A8 of our revised manuscript, we explore more rank values varying from 1 to 50. We can see that the performance of LoRA increases with the larger rank values, but the difference between $r = 10$ and $r = 50$ is insignificant. In contrast, the efficiency of LoRA will drop significantly with a rank larger than 10. In order to strike a balance between performance and efficiency, we adopt the rank value of 10 as the default value in this work.

Table A8: Ablation study on performance of LoRA with different rank values. Experiment settings follow Tab. 1 and Tab. 4.

| | Accuracy | | | Efficiency | | | |
| --- | --- | --- | --- | --- | --- | --- | --- |
| | **Full-Data** | | | | **Train-Time Efficiency** | | |
| | FGVC | VTAB | Others | Param. # | Memory | Time | Throughput |
| Number of tasks | 5 | 9 | 5 | - | - | - | - |
| FULL-FINETUNE | 91.43 | 91.97 | 93.91 | 304.33 | 41.5 | 520 | 79.58 |
| LINEAR-PROBE | 82.23 | 78.90 | 87.81 | 0.01 | 9.7 | 121 | 79.64 |
| LoRA-1 | 84.43 | 88.21 | 90.07 | 0.04 | 10.43 | 139 | 79.43 |
| LoRA-10 | 86.87 | 89.81 | 91.45 | 1.00 | 33.1 | 363 | 79.43 |
| LoRA-20 | 86.93 | 90.23 | 91.35 | 4.38 | 33.1 | 443 | 79.43 |
| LoRA-50 | 87.23 | 90.41 | 91.97 | 12.22 | 57.2 | 589 | 79.43 |
| AP | 85.30 | 90.25 | 91.09 | 0.16 | 31.6 | 262 | 79.43 |

**Application of AP to multiple layers.** We implement AP with multiple layers, and we show the results in Tab. A9 in the revision. Our findings indicate that the layer addition of AP does not yield significant improvements in performance. This observation is significant as it suggests that applying AP to a single, carefully selected layer can achieve comparable performance to more extensive applications. This underscores the efficiency of AP, affirming its value in settings where computational resources are a concern.

Table A9: Ablation study on the number of layers installed with AP. In particular, for AP-3 and AP-5, AP are installed on the input of the first 3 and 5 blocks of the pretrained ViT-L. Other experiment settings follow Tab. 1, and Tab. 2.

| | Accuracy | | | Efficiency | | | |
|---|---|---|---|---|---|---|---|
| | **Full-Data** | | | **Train-Time Efficiency** | | | |
| | FGVC | VTAB | Others | Param. # | Memory | Time | Throughput |
| Number of tasks | 5 | 9 | 5 | - | - | - | - |
| FULL-FINETUNE | 91.43 | 91.97 | 93.91 | 304.33 | 41.5 | 520 | 79.58 |
| LINEAR-PROBE | 82.23 | 78.90 | 87.81 | 0.01 | 9.7 | 121 | 79.64 |
| BIAS | 85.32 | 89.84 | 90.41 | 0.29 | 32.9 | 297 | 79.48 |
| LoRA | 86.87 | 89.81 | 91.45 | 1.00 | 33.1 | 363 | 79.43 |
| VPT | 86.05 | 89.97 | 90.64 | 1.24 | 38.6 | 397 | 72.84 |
| ADAPTER | 87.06 | 89.44 | 91.21 | 2.17 | 32.4 | 357 | 63.39 |
| ADAPTERFORMER | **89.18** | 90.69 | **92.08** | 0.65 | 32.3 | 289 | 23.69 |
| AP-3 | 85.41 | 90.38 | 91.21 | 0.46 | 47.8 | 297 | 79.43 |
| AP-5 | 85.49 | 90.49 | 91.31 | 0.76 | 69.7 | 348 | 79.43 |
| AP | 85.30 | 90.25 | 91.09 | 0.16 | 31.6 | 262 | 79.43 |

## C  THEORETICAL DETAILS

### C.1  MODEL ARCHITECTURE

We define the general definition of the model architecture CNN, ViT in this section.

**CNN**: We follow the architecture of ResNet (), which stacks multiple residual blocks plus an input and an output layer. Each residual block includes several convolutional layers and a skip connection. For the input $z_{\text{in}}^{(l)}$ to the $l$-th convolutional layer, where $l \in [L]$, the output $z_{\text{out}}^{(l)}$ can be computed as

$$z^{(l)} = \text{Conv}(z_{\text{in}}^{(l)}; W_1^{(l)}), \; z_{\text{out}}^{(l)} = \text{relu}(\text{BN}(z^{(l)})) \tag{A1}$$

where $z_{\text{in}}^{(0)} = x$. $\text{Conv}(\cdot)$ and BN denote the Convolution operation and the Batch Normalization, respectively. The output $\hat{y} = \text{FC}(\text{Pooling}(z_{\text{out}}^{(L)}))$, where $\text{FC}(\cdot)$ denotes fully-connected layer.

**ViT**: The architecture of Vision Transformer is defined in (). For the input $z_{\text{in}}^{(l)}$ to the $l$-th Transformer layer, we first let $z^{(l)} = z_{\text{in}}^{(l)}$. Then, the output $z_{\text{out}}^{(l)}$ can be computed as

$$z^{(l)} = \text{MSA}(\text{LN}(z^{(l)})) + z^{(l)}, \; z_{\text{out}}^{(l)} = \text{MLP}(\text{LN}(z^{(l)})) + z^{(l)}, \tag{A2}$$

where $z_{\text{in}}^{(0)} = x$. $\text{MSA}(\cdot)$ and $\text{LN}(\cdot)$ denote the Multi-Head Self-attention and Layer Normalization, respectively. For an $L$-layer ViT, the output $\hat{y} = \text{Out}(H_{\text{out}}^{(L)})$, where $\text{Out}(\cdot)$ denotes the output layer.

### C.2  PROPOSITION 1 AND ITS PROOF

We first provide a full definition of NORM-TUNE.

NORM-TUNE is a method where only the Batch Normalization layers for CNNs or Layer Normalization for ViTs are trainable. Consider a batch of the $l$-th-layer features $z_1^{(l)}, z_2^{(l)}, \cdots, z_B^{(l)}$ defined in (A1) and (A2), where $z_b^{(l)} = [z_{b,\cdot,1}^{(l)}, z_{b,\cdot,2}^{(l)}, \cdots, z_{b,\cdot,P'}^{(l)}] = \in \mathbb{R}^{D' \times P'}$, $z_{b,\cdot,p}^{(l)} \in \mathbb{R}^{D'}$ for $b \in [B]$ and $p \in [P']$. $B$ is the batch size, $D'$ denotes the number of channels or token dimension, and $P'$ denotes the size of the feature map or token length. We can formulate the Normalization on $h_{b,d,p}^{(l)}$, the $d$-th dimension of $h_{b,\cdot,p}^{(l)}$, as follows.

$$\textbf{BN}: \mu_d = \sum_{b=1}^{B} \sum_{p=1}^{P'} \frac{z_{b,d,p}^{(l)}}{BP'}, \; \sigma_d^2 = \sum_{b=1}^{B} \sum_{p=1}^{P'} \frac{(z_{b,d,p}^{(l)} - \mu_d)^2}{BP'}, \; \text{BN}(z_{b,d,p}^{(l)}) = \gamma_d \frac{z_{b,d,p}^{(l)} - \mu_d}{\sigma_d} + \beta_d,$$

$$\textbf{LN}: \mu_{b,p} = \sum_{d=1}^{D'} \frac{z_{b,d,p}^{(l)}}{D'}, \; \sigma_{b,p}^2 = \sum_{d=1}^{D'} \frac{(z_{b,d,p}^{(l)} - \mu_{b,p})^2}{D'}, \; \text{LN}(z_{b,d,p}^{(l)}) = \gamma_d \frac{z_{b,d,p}^{(l)} - \mu_{b,p}}{\sigma_{b,p}} + \beta_d,$$

$$\tag{A3}$$

where $\gamma_d$, $\beta_d$ are trainable parameters for $d \in [D']$. Then, we present a full statement of Proposition 1.

**Proposition 1** *Without the assumption that the input to the batch (or layer) normalization layer has zero mean and unit variance for each dimension (or token), we have the following conclusion:*

AP *on the $l$-th layer is the same as* NORM-TUNE *on the $l$-th layer, if*

- *for CNNs, $\gamma_d/\sigma_d = 1$, and all $\boldsymbol{\delta}_p$'s added to $\boldsymbol{z}_b^{(l)}$ are the same as $\boldsymbol{\delta}$, $\beta_d = \boldsymbol{w}_d^{(l)}\boldsymbol{\delta}_* + \boldsymbol{\mu}_d$ for all $d \in [D']$, where $\boldsymbol{\delta}_* = \boldsymbol{\delta}_i^{(l)}$ for $i \in [P']$;*

- *for ViTs, $\gamma_d/\sigma_{b,p} = 1$, and $\mu_{b,p}$'s are the same as $\mu_p$, $p \in [P']$ among all $b \in [B]$ for ViTs, $\beta_d = \delta_{p,d}^{(l)} + \mu_p$ for all $d \in [D']$, $p \in [P']$.*

**Proof:**

For BN, note that

$$\text{BN}(z_{b,d,p}^{(l)}) = \gamma_d \frac{z_{b,d,p}^{(l)} - \mu_d}{\sigma_d} + \beta_d = \frac{\gamma_d}{\sigma_d} z_{b,d,p}^{(l)} + \beta_d - \frac{\mu_d \gamma_d}{\sigma_d} \tag{A4}$$

where

$$z_{b,d,p}^{(l)} = \boldsymbol{w}_d^{(l)} \boldsymbol{z}_{b,\cdot,p}^{(l-1)}, \quad \boldsymbol{z}_{b,\cdot,p}^{(l-1)} = \boldsymbol{x}_{b,\cdot,p} \tag{A5}$$

When adding the prompt $\boldsymbol{\delta}_p^{(l)}$, we have the output

$$\boldsymbol{w}_d^{(l)}(\boldsymbol{z}_{b,\cdot,p}^{(l-1)} + \boldsymbol{\delta}_p^{(l)}) \tag{A6}$$

We then need the equation

$$\frac{\gamma_d}{\sigma_d} z_{b,d,p}^{(l)} + \beta_d - \frac{\mu_d \gamma_d}{\sigma_d} = \boldsymbol{w}_d^{(l)}(\boldsymbol{z}_{b,\cdot,p}^{(l-1)} + \boldsymbol{\delta}_p^{(l)}) \tag{A7}$$

Given $\gamma_d/\sigma_d = 1$, we have

$$\beta_d = \boldsymbol{w}_d^{(l)} \boldsymbol{\delta}_p^{(l)} + \mu_d \tag{A8}$$

Suppose that $\mu_d = 0$ for $d \in [D']$ and $\boldsymbol{\delta}_p^{(l)} = \boldsymbol{\delta}_*$ for $p \in [P']$, we can obtain

$$\beta_d = \boldsymbol{w}_d^{(l)} \boldsymbol{\delta}_* \tag{A9}$$

For LN, we need

$$\text{LN}(z_{b,d,p}^{(l)}) = \gamma_d \frac{z_{b,d,p}^{(l)} - \mu_{b,p}}{\sigma_{b,p}} + \beta_d = \frac{\gamma_d}{\sigma_{b,p}} z_{b,d,p}^{(l)} + \beta_d - \frac{\gamma_d \mu_{b,p}}{\sigma_{b,p}} = z_{b,d,p}^{(l)} + \delta_{p,d}^{(l)} \tag{A10}$$

Given $\gamma_d/\sigma_{b,p} = 1$ and $\boldsymbol{\mu}_{b,p} = \boldsymbol{\mu}_p$ for $b \in [B]$, we have

$$\beta_d = \delta_{p,d}^{(l)} + \mu_p \tag{A11}$$

Suppose that $\mu_p = 0$, $p \in [P']$ and let $\boldsymbol{\delta}_p^{(l)} = \boldsymbol{\delta}_*$, $p \in [P']$, we can obtain

$$\boldsymbol{\beta} = \boldsymbol{\delta}_* \tag{A12}$$

## C.3 PROOF OF LEMMA 1

Before we provide the proof, we state the formulation of a single-head and two-layer ViT, the full assumption on the data model, and the pretrained model in detail.

Let $\boldsymbol{x}_{n(\cdot,j)}$ be the $j$-th patch/token of $\boldsymbol{x}_n$, $j \in [P]$. The corresponding 1-st-layer output is $\boldsymbol{z}_{n(\cdot,j)}$. Denote the $j$-th patch/token of $\boldsymbol{x}_n$ or $\boldsymbol{z}_n$ after introducing the AP, $\boldsymbol{\delta}^{(h)}$, as $\boldsymbol{x}_n[\boldsymbol{\delta}_j^{(h)}]$ and $\boldsymbol{z}_n[\boldsymbol{\delta}_j^{(h)}] = (\boldsymbol{z}_n[\boldsymbol{\delta}_1^{(h)}], \cdots, \boldsymbol{z}_n[\boldsymbol{\delta}_P^{(h)}])$, respectively.

Following (Dosovitskiy et al., 2020), we consider a single-head self-attention parameterized by $\boldsymbol{W}_Q^{(l)}$, $\boldsymbol{W}_K^{(l)}$, and $\boldsymbol{W}_V^{(l)}$ in the $l$-th layer. The shapes of these matrices are $m$ by $d$ if $l = 1$ and $m$ by $m$ if $l = 2$. Denote $\boldsymbol{W}^{(l)} = \boldsymbol{W}_K^{(l)}{}^\top \boldsymbol{W}_Q^{(l)}$, $l = 1, 2$. The MLP layer is a two-layer perceptron with $m \times m$-dimensional parameters $\boldsymbol{W}_O^{(l)}$, $\boldsymbol{W}_U^{(l)}$, and Relu activation. The output layer is a fully-connected layer with $\boldsymbol{a}_1, \cdots, \boldsymbol{a}_P$ where $\boldsymbol{a}_l \in \mathbb{R}^m$. Then, a two-layer ViT can be written as

$$f_{\boldsymbol{\theta}}(\boldsymbol{x}_n, \boldsymbol{\delta}^{(h)}) = \sum_{k=1}^P \boldsymbol{a}_k^\top \boldsymbol{W}_U^{(2)} \text{Relu}(\boldsymbol{W}_O^{(2)} \boldsymbol{W}_V^{(2)} \boldsymbol{z}_n[\boldsymbol{\delta}^{(h)}] \text{softmax}(\boldsymbol{z}_n[\boldsymbol{\delta}^{(h)}]^\top \boldsymbol{W}^{(2)} \boldsymbol{z}_n[\boldsymbol{\delta}_k^{(h)}])),$$

$$\boldsymbol{z}_n[\boldsymbol{\delta}_k^{(1)}] = \boldsymbol{W}_U^{(1)} \text{Relu}(\sum_{s=1}^P \boldsymbol{W}_O^{(1)} \boldsymbol{W}_V^{(1)} \boldsymbol{x}_n[\boldsymbol{\delta}_s^{(h)}] \text{softmax}(\boldsymbol{x}_n[\boldsymbol{\delta}_s^{(h)}]^\top \boldsymbol{W}^{(1)} \boldsymbol{x}_n[\boldsymbol{\delta}_k^{(h)}])),$$

(A13)

The AP is restated as

$$\begin{cases} \boldsymbol{x}_n[\boldsymbol{\delta}_j^{(h)}] = \boldsymbol{x}_{n(\cdot,j)} + \boldsymbol{\delta}_j^{(h)}, \boldsymbol{z}_n[\boldsymbol{\delta}_j^{(h)}] \text{ as defined in (A13)}, & \text{if } h = 1, \\ \boldsymbol{x}_n[\boldsymbol{\delta}_j^{(h)}] = \boldsymbol{x}_{n(\cdot,j)}, \boldsymbol{z}_n[\boldsymbol{\delta}_j^{(h)}] = \boldsymbol{z}_{n(\cdot,j)} + \boldsymbol{\delta}_j^{(h)}, & \text{if } h = 2, \end{cases}$$

(A14)

We use Hinge loss $\ell(\boldsymbol{x}_n.y_n) = \max\{0, 1/P - y_n f_{\boldsymbol{\theta}}(\boldsymbol{x}_n, \boldsymbol{\delta}^{(h)})\}$ as the loss function.

**Data model** The patch/token $\boldsymbol{x}_{n(\cdot,j)}$ is a noisy version of patterns, i.e., $\boldsymbol{x}_{n(\cdot,j)} = \boldsymbol{v}_l + \epsilon_j^n$, where $\boldsymbol{v}_l$, $l = 1, 2, 3, 4$ is a pattern and $\epsilon_j^n \sim \mathcal{N}(0, \sigma^2)$ is a Gaussian noise, $\sigma \le O(1/P)$. $\boldsymbol{v}_1, \boldsymbol{v}_2, \boldsymbol{v}_3, \boldsymbol{v}_4$ are all unit norm and orthogonal to each other except the pairs of $\boldsymbol{v}_3$ and $\boldsymbol{v}_4$. $\boldsymbol{v}_3^\top \boldsymbol{v}_4 = \zeta \in (-1, 0)$. In each sample $\boldsymbol{x}_n$, only one patch/token $\boldsymbol{x}_{n(\cdot,j)}$ corresponds to either $\boldsymbol{v}_1$ or $\boldsymbol{v}_2$, while other $P - 1$ patches/tokens correspond to either $\boldsymbol{v}_3$ or $\boldsymbol{v}_4$. $\boldsymbol{v}_1, \boldsymbol{v}_2$ are called discriminative patterns that decide the label. $\boldsymbol{v}_3, \boldsymbol{v}_4$ are non-discriminative patterns that work as the image background. For instance, if one patch is the noisy version of $\boldsymbol{v}_1$ ($\boldsymbol{v}_2$), then $y^n = 1$ ($y^n = -1$).

**Pretrained model** The pretraining stage is assumed to learn a task where all patterns $\{\boldsymbol{v}_1, \boldsymbol{v}_2, \boldsymbol{v}_3, \boldsymbol{v}_4\}$ are key features, where each data contains two types of patterns. The label is determined by the number of $\boldsymbol{v}_1$ or $\boldsymbol{v}_3$ compared with the number of $\boldsymbol{v}_2$ or $\boldsymbol{v}_4$. Inspired by the finding that some trained "lucky" hidden neurons represent discriminative features from existing theoretical works () on VITs, we accordingly set the neurons of feed-forward-networks $\boldsymbol{W}_O^{(i)}$ in (A13), $i = 1, 2$ as pattern representations of that layer and ignore "unlucky" neurons, which has a trivial effect on the output. To be more specific, for the 1st layer, we set a $1/4$ fraction of neurons to be $\boldsymbol{v}_i$, $i = 1, 2, 3, 4$, and for the 2nd layer, we set a $1/4$ fraction of neurons to be $\boldsymbol{e}_i$, $i = 1, 2, 3, 4$, i.e., the 2nd-layer pattern representations. $\boldsymbol{a}_i$ for CNNs and $\boldsymbol{a}_{l(i)}$ equal $1/(mP)$ for neurons of $\boldsymbol{e}_1$ and $\boldsymbol{e}_3$, and they equal $-1/(mP)$ for neurons of $\boldsymbol{e}_2$ and $\boldsymbol{e}_4$. For ViTs, we follow the orthogonal embedding assumption in (Oymak et al., 2023; Li et al., 2023a; Zhang et al., 2023b) and set $\boldsymbol{W}_Q^{(1)} = \beta_1 \boldsymbol{I}$, $\boldsymbol{W}_K^{(1)} = \beta_1 \boldsymbol{P}_x^{(1)}$, $\boldsymbol{W}_Q^{(2)} = \beta_2 \boldsymbol{I}$, $\boldsymbol{W}_K^{(2)} = \beta_2 \boldsymbol{P}_x^{(2)}$, $\boldsymbol{W}_V^{(1)} = \boldsymbol{P}_x^{(1)}$, $\boldsymbol{W}_V^{(2)} = \boldsymbol{P}_x^{(2)}$ for simplicity, where $\beta_1 = \Theta(1)$, $\beta_2 = \Theta(1)$, $\boldsymbol{I}$ is the identity matrix, and $\boldsymbol{P}_x^{(1)}$ and $\boldsymbol{P}_x^{(2)}$ are permutation matrices.

Then, we present the proof of Lemma 1.

**Proof:**

Without loss of generality, we focus on studying the data where $\boldsymbol{v}_1$ is the discriminative pattern, and $\boldsymbol{v}_4$ is the non-discriminative pattern.

For ViTs, note that the permutation matrix $\boldsymbol{P}_x^{(1)}$ changes the location of the pattern $\boldsymbol{v}_1$ to another place with a distance of at least $d_A$. By computing the feature correlation for the pattern $\boldsymbol{v}_1$, we have

$$\beta_1^2 > 0,$$

(A15)

which means the the pattern $\boldsymbol{v}_1$ has the largest correlation with $\boldsymbol{v}_1$. Hence, the pattern of $\boldsymbol{v}_1$ is a global feature. For the feature correlation of the pattern $\boldsymbol{v}_4$, we have

$$\beta_1^2 > 0,$$

(A16)

which means the the pattern $\boldsymbol{v}_4$ has the largest correlation with $\boldsymbol{v}_4$. Hence, the pattern of $\boldsymbol{v}_4$ is a global feature because the distance between two $\boldsymbol{v}_4$ patterns is at most 1. Since that there will be one

$v_4$ token corresponding to a $v_1$ token after the permutation, there will be a contribution of distance 1 to the average distance. The average attention distance of the first layer is

$$\frac{1}{P}\sum_{i=1}^{P}|i - \arg\max_{j\in[P]}\langle k_j, q_i\rangle| = \frac{1+d_A}{P} \tag{A17}$$

After the first layer, the feature of the $v_1$ token becomes

$$\frac{e^{\beta_1^2}}{e^{\beta_1^2}+P-1}v_1 + \frac{P-1}{e^{\beta_1^2}+P-1}v_4 := \lambda_1 v_1 + (1-\lambda_1)v_4, \tag{A18}$$

while the feature of the $v_4$ token becomes

$$\frac{1}{(P-1)e^{\beta_1^2}+1}v_1 + \frac{(P-1)e^{\beta_1^2}}{(P-1)e^{\beta_1^2}+1}v_4 := \lambda_2 v_1 + (1-\lambda_2)v_4, \tag{A19}$$

Here $1/2 > \lambda_1 > \lambda_2 > 0$. Therefore, we have

$$(\lambda_1 v_1 + (1-\lambda_1)v_4)^\top(\lambda_1 v_1 + (1-\lambda_1)v_4 - \lambda_2 v_1 - (1-\lambda_2)v_4)$$
$$=(2\lambda_1 - 1)(\lambda_1 - \lambda_2) < 0 \tag{A20}$$

$$(\lambda_2 v_1 + (1-\lambda_2)v_4)^\top(\lambda_2 v_1 + (1-\lambda_2)v_4 - \lambda_1 v_1 - (1-\lambda_1)v_4)$$
$$=(2\lambda_2 - 1)(\lambda_2 - \lambda_1) > 0 \tag{A21}$$

Therefore, the feature from the token of $v_4$ has the largest correlation with the token of both $v_1$ and $v_4$. Since there exists a $v_4$ token close to $v_1$ token with a distance of at most 1, we have that both $v_1$ and $v_4$ tokens become local features. Then, the average attention distance of the second layer is

$$\frac{1}{P}\sum_{i=1}^{P}|i - \arg\max_{j\in[P]}\langle k_j, q_i\rangle| = \frac{1}{P} \tag{A22}$$

## C.4 Proof of Theorem 1

We first present two lemmas. One can observe that Theorem 1 is a combination of these two lemmas. Therefore, the proof of Theorem 1 is exactly the same as the proof of these two lemmas.

**Lemma 2** *For a two-layer single-head Transformer*

$$f_\theta(x_n, \delta) = \sum_{l=1}^{P}\sum_{i=1}^{m} a_{l(i)}^\top Relu(\sum_{j=1}^{P} W_{O_{2(i,\cdot)}} W_{V_2}(z_{n(\cdot,j)} + \delta_j^{(h)}) \tag{A23}$$
$$\cdot softmax((z_{n(\cdot,j)} + \delta_j^{(h)})^\top W_{K_2}^\top W_{Q_2}(z_{n(\cdot,l)} + \delta_l^{(h)})))$$

*where*

$$z_{n(\cdot,j)} = Relu(\sum_{s=1}^{P} W_{O_1} W_{V_1} x_{n(\cdot,s)} softmax(x_{n(\cdot,s)}^\top W_{K_1}^\top W_{Q_1} x_{n(\cdot,j)})) \tag{A24}$$

*as long as the batch size and the required number of iterations satisfy*

$$B \geq \Omega(1), \quad T = \frac{\eta^{-1}P^2\log P}{(1-\sigma)^{-1}}, \tag{A25}$$

*where $\sigma \leq \Theta(P^{-1})$, training $\delta^{(h)}$, $h = 2$ with SGD returns a model with zero generalization error.*

**Lemma 3** *For a two-layer single-head Transformer*

$$f_\theta(x_n, \delta) = \sum_{l=1}^{P}\sum_{i=1}^{m} a_{l(i)}^\top Relu(\sum_{j=1}^{P} W_{O_{2(i,\cdot)}} W_{V_2} z_{n(\cdot,j)} softmax(z_{n(\cdot,j)}^\top W_{K_2}^\top W_{Q_2} z_{n(\cdot,l)}))$$
$$\tag{A26}$$

*where*

$$z_{n(\cdot,j)} = Relu(\sum_{s=1}^{P} W_{O_1} W_{V_1}(x_{n(\cdot,s)} + \delta_s^{(h)})softmax((x_{n(\cdot,s)} + \delta_s^{(h)})^\top W_{K_1}^\top W_{Q_1}(x_{n(\cdot,j)} + \delta_j^{(h)}))) \tag{A27}$$

*as long as the batch size and the required number of iterations satisfy*

$$B \geq \Omega(1), \quad T = \frac{\eta^{-1}P}{(1 - P\sigma)^{-1}(1 + \gamma)}, \tag{A28}$$

*where $\sigma \leq O(P^{-1})$, training $\delta^{(h)}$, $h = 1$ with SGD returns a model with zero generalization error, where $\gamma := v_3^\top v_4 \in (-1, 0)$.*

### C.4.1 PROOF OF LEMMA 2

**Proof:**

For $h = 2$,

$$f_\theta(x_n, \delta^{(h)}) = \sum_{l=1}^{P} \sum_{i=1}^{m} a_{l(i)}^\top Relu(\sum_{s=1}^{P} W_{O_{(i,\cdot)}} W_V(z_{n(\cdot,s)} + \delta_s^{(h)}) \tag{A29}$$
$$\cdot softmax((z_{n(\cdot,s)} + \delta_s^{(h)})^\top W_K^\top W_Q(z_{n(\cdot,s)} + \delta_l^{(h)}))),$$

we have $W_K = \beta_2 \cdot P_x$, $W_Q = \beta_2 \cdot I$, and $W_V = P_x$ where $\beta_2 = \Theta(1)$. To avoid multiple superscripts, we use $\delta$ to denote $\delta^{(h)}$ since that $h$ is fixed in this proof. We use $\delta^{(t)}$ to denote the update of $\delta$ at $t$-th iteration. Then,

$$\frac{\partial f_\theta(x_n, \delta)}{\partial \delta_j}$$
$$= \sum_{l=1}^{P} \sum_{i=1}^{m} a_{l(i)} \mathbb{1}[\sum_{s=1}^{P} W_{O_{(i,\cdot)}}(z_{n(\cdot,P_{s,2})} + \delta_{P_{s,2}})softmax((z_{n(\cdot,P_{s,2})} + \delta_{P_{s,2}})^\top(z_{n(\cdot,s)}$$
$$+ \delta_l)) \geq 0] \cdot \Big(softmax((z_{n(\cdot,P_{s,2})} + \delta_{P_{s,2}})^\top(z_{n(\cdot,s)} + \delta_l))W_{O_{(i,\cdot)}} \tag{A30}$$
$$+ \mathbb{1}[j \neq l]W_{O_{(i,\cdot)}}(z_{n(\cdot,j)} + \delta_j) \cdot (z_{n(\cdot,j)} + \delta_l) \cdot (-softmax(\beta_2^2(z_{n(\cdot,j)} + \delta_j)^\top$$
$$\cdot (z_{n(\cdot,l)} + \delta_l)))softmax(\beta_2^2(z_{n(\cdot,l)} + \delta_l)^\top(z_{n(\cdot,l)} + \delta_l))$$
$$+ \mathbb{1}[j = l]W_{O_{(i,\cdot)}}(z_{n(\cdot,l)} + \delta_l)softmax(\beta_2^2(z_{n(\cdot,l)} + \delta_l)^\top(z_{n(\cdot,l)} + \delta_l))$$
$$\cdot (1 - softmax(\beta_2^2(z_{n(\cdot,l)} + \delta_l)^\top(z_{n(\cdot,j)} + \delta_l)))(z_{n(\cdot,l)} + \delta_l)$$

Let $t = 0$. For $y^n = +1$, Note that if $z_n = [e_3, e_3, \cdots, e_3, e_1, e_3, \cdots, e_3]$ without noise, the loss is 0. Hence, we compute the loss from $z_n = [e_4, e_4, \cdots, e_4, e_1, e_4, \cdots, e_4]$.

$$\mathbb{E}[\mathbb{1}[\sum_{s=1}^{P} W_{O_{(i,\cdot)}}(x_{n(\cdot,s)} + \delta_s^{(t)})softmax(\beta_2^2(z_{n(\cdot,P_{s,2})} + \delta_{P_{s,2}}^{(t)})^\top(z_{n(\cdot,l)} + \delta_l^{(t)})) \geq 0]$$
$$= Pr(\sum_{s=1}^{L} W_{O_{(i,\cdot)}}(z_{n(\cdot,P_{s,2})} + \delta_{P_{s,2}}^{(t)})softmax(\beta_2^2(z_{n(\cdot,P_{s,2})} + \delta_{P_{s,2}}^{(t)})^\top(z_{n(\cdot,l)} + \delta_l^{(t)})) \geq 0) \tag{A31}$$

for $W_{O_{(i,\cdot)}} = e_1$ or $e_4$. We can finally show that with a high probability, the above indicator is close to 1. Meanwhile, for $W_{O_{(i,\cdot)}} = e_2$ or $e_3$, the indicator equals 0 or 1 with half probability when $t = 0$. Consider that $z_{n(\cdot,j)}$ comes from $v_4$, which means $z_{n(\cdot,j)}$ is close to $v_4$ by a noisy term. In this case, if $z_{n(\cdot,l)}$ comes from $v_1$,

$$softmax(\beta_2^2(z_{n(\cdot,l)} + \delta_l^{(t)})^\top(z_{n(\cdot,l)} + \delta_l^{(t)})) \geq \frac{1}{P} \tag{A32}$$

$$softmax(\beta_2^2(z_{n(\cdot,j)} + \delta_j)^\top(z_{n(\cdot,l)} + \delta_l^{(t)})) = \Theta(\frac{1}{P}) \tag{A33}$$

If $\boldsymbol{z}_{n(\cdot,l)}$ comes from $\boldsymbol{v}_4$, then

$$\text{softmax}(\beta_2^2(\boldsymbol{z}_{n(\cdot,l)} + \boldsymbol{\delta}_l^{(t)})^\top (\boldsymbol{z}_{n(\cdot,l)} + \boldsymbol{\delta}_l^{(t)})) \geq \frac{1}{P} \tag{A34}$$

$$\text{softmax}(\beta_2^2(\boldsymbol{z}_{n(\cdot,j)} + \boldsymbol{\delta}_j^{(t)})^\top (\boldsymbol{z}_{n(\cdot,l)} + \boldsymbol{\delta}_l^{(t)})) = \Theta(\frac{1}{P}) \tag{A35}$$

Then we consider that $\boldsymbol{z}_{n(\cdot,j)}$ comes from $\boldsymbol{e}_1$. In this case, if $\boldsymbol{z}_{n(\cdot,l)}$ comes from $\boldsymbol{v}_1$, then

$$\text{softmax}(\beta_2^2(\boldsymbol{z}_{n(\cdot,j)} + \boldsymbol{\delta}_j^{(t)})^\top (\boldsymbol{z}_{n(\cdot,l)} + \boldsymbol{\delta}_l^{(t)})) \geq \frac{1}{P} \tag{A36}$$

If $\boldsymbol{z}_{n(\cdot,l)}$ comes from $\boldsymbol{v}_4$,

$$\text{softmax}(\beta_2^2(\boldsymbol{z}_{n(\cdot,j)} + \boldsymbol{\delta}_j^{(t)})^\top (\boldsymbol{z}_{n(\cdot,l)} + \boldsymbol{\delta}_l^{(t)})) \leq \frac{1}{P} \tag{A37}$$

Therefore, if $\boldsymbol{z}_{n(\cdot,j)}$ comes from $\boldsymbol{v}_1$,

$$\frac{\partial f_{\boldsymbol{\theta}}(\boldsymbol{x}_n, \boldsymbol{\delta}^{(t)})}{\partial \boldsymbol{\delta}_j^{(t)}} = \frac{1}{4P}\lambda \boldsymbol{e}_1 + \Theta(\frac{1}{P})(-\boldsymbol{e}_2 + \boldsymbol{e}_3 - \boldsymbol{e}_4), \tag{A38}$$

and if $\boldsymbol{z}_{n(\cdot,j)}$ comes from $\boldsymbol{v}_4$,

$$\frac{\partial f_{\boldsymbol{\theta}}(\boldsymbol{x}_n, \boldsymbol{\delta}^{(t)})}{\partial \boldsymbol{\delta}_j^{(t)}} = -\frac{1}{4P}\lambda \boldsymbol{e}_4 + \Theta(\frac{1}{P})(-\boldsymbol{e}_2 + \boldsymbol{e}_3 + \boldsymbol{e}_1), \tag{A39}$$

where $\lambda = \mu = \Theta(1)$. The last terms in (A38) and (A39) come from the indicators from other $\boldsymbol{W}_O$ neurons, which may become 1 because of feature noises. Note that when $t \geq 2$, since the data which contains $\boldsymbol{e}_2$ and $\boldsymbol{e}_3$ would similarly contribute to the overall gradient, there will be a close amount of $\boldsymbol{e}_1$ and $\boldsymbol{e}_2$ in $\boldsymbol{\delta}_j^{(t)}$ and a close amount of $\boldsymbol{e}_3$ and $\boldsymbol{e}_4$ in $\boldsymbol{\delta}_j^{(t)}$. Hence, when $k\mu < \Theta(1)$,

$$
\begin{aligned}
\mathbb{E}[\boldsymbol{\delta}_j^{(t)}] &= \mathbb{E}[\boldsymbol{\delta}_j^{(0)}] - \mathbb{E}[\eta \sum_{b=1}^{t} \frac{1}{B} \sum_{n \in \mathcal{B}_b} \frac{\partial}{\partial \boldsymbol{\delta}_j} \ell(f_{\boldsymbol{\theta}}(\boldsymbol{x}_n, \boldsymbol{\delta}^{(b)}), y_n)] \\
&= \eta t \frac{1}{4P}(\lambda \boldsymbol{e}_1 + \lambda \boldsymbol{e}_2 - \mu \boldsymbol{e}_3 - \mu \boldsymbol{e}_4) \\
&= k(\lambda \boldsymbol{e}_1 + \lambda \boldsymbol{e}_2 - \mu \boldsymbol{e}_3 - \mu \boldsymbol{e}_4),
\end{aligned} \tag{A40}
$$

$$\boldsymbol{\delta}_j^{(t)} = \mathbb{E}[\boldsymbol{\delta}_j^{(t)}] + \frac{\eta t}{L}\sqrt{\frac{\log Bt}{Bt}}(\pm \boldsymbol{e}_1 \pm \boldsymbol{e}_2 \pm \boldsymbol{e}_3 \pm \boldsymbol{e}_4) \tag{A41}$$

where $\lambda \geq \Theta(1) \cdot (1 - \sigma P)$, $\mu \geq \Theta(1) \cdot (1 - \sigma P)$ for $t \geq 2$. The term $(1 - \sigma P)$ comes from that for $\boldsymbol{W}_{O(i,\cdot)} = \boldsymbol{e}_1$ or $\boldsymbol{e}_4$,

$$\mathbb{E}[\mathbb{1}[\sum_{s=1}^{P} \boldsymbol{W}_{O(i,\cdot)}(\boldsymbol{z}_{n(\cdot,P_{s,2})} + \boldsymbol{\delta}_{P_{s,2}}^{(t)})\text{softmax}(\beta_2^2(\boldsymbol{z}_{n(\cdot,P_{s,2})} + \boldsymbol{\delta}_{P_{s,2}}^{(t)})^\top (\boldsymbol{z}_{n(\cdot,l)} + \boldsymbol{\delta}_l^{(t)})) \geq 0]$$

$$\geq 1 - e^{\frac{(Bt)^2}{\sigma^2 P^2}} \geq 1 - \sigma P \tag{A42}$$

given $B \geq \Theta(1)$ by Hoeffding inequality. When $k\mu \geq \Theta(1)$, for $\boldsymbol{z}_n = [\boldsymbol{e}_4, \boldsymbol{e}_4, \cdots, \boldsymbol{e}_4, \boldsymbol{e}_1, \boldsymbol{e}_4, \cdots, \boldsymbol{e}_4]$,

$$\boldsymbol{z}_{n(\cdot,j)} + \boldsymbol{\delta}_j^{(t)} = k\lambda(\boldsymbol{e}_1 + \boldsymbol{e}_2) - k\mu\boldsymbol{e}_3 + (1 - k\mu)\boldsymbol{e}_4 \tag{A43}$$

for $\boldsymbol{z}_{n(\cdot,j)}$ from $\boldsymbol{v}_4$. Then,

$$\mathbb{E}[\mathbb{1}[\sum_{s=1}^{P} \boldsymbol{e}_1(\boldsymbol{z}_{n(\cdot,P_{s,2})} + \boldsymbol{\delta}_{P_{s,2}}^{(t)})\text{softmax}(\beta_2^2(\boldsymbol{z}_{n(\cdot,P_{s,2})} + \boldsymbol{\delta}_{P_{s,2}}^{(t)})^\top (\boldsymbol{z}_{n(\cdot,l)} + \boldsymbol{\delta}_l^{(t)}))]] \geq 1 - e^{\frac{(Bt)^2}{\sigma^2}} \geq 1 - \sigma \tag{A44}$$

$$\Pr(\sum_{s=1}^{P} \boldsymbol{e}_4(\boldsymbol{z}_{n(\cdot,P_{s,2})} + \boldsymbol{\delta}_{P_{s,2}}^{(t)})\text{softmax}(\beta_2^2(\boldsymbol{z}_{n(\cdot,P_{s,2})} + \boldsymbol{\delta}_{P_{s,2}}^{(t)})^\top(\boldsymbol{z}_{n(\cdot,l)} + \boldsymbol{\delta}_l^{(t)}))) \leq e^{-\frac{1}{\sigma^2}} \leq e^{-P^2}$$
(A45)

Hence, with a probability at least $1 - e^{-P^2}$, no patches is activated by $\boldsymbol{e}_4$. For $\boldsymbol{z}_{n(\cdot,k)}$ from $\boldsymbol{v}_1$ and $\boldsymbol{z}_{n(\cdot,j)}$ from $\boldsymbol{v}_4$, we have

$$\text{softmax}((\boldsymbol{z}_{n(\cdot,k)} + \boldsymbol{\delta}_k^{(t)})^\top(\boldsymbol{z}_{n(\cdot,k)} + \boldsymbol{\delta}_k^{(t)})) \geq \frac{1}{P}$$
(A46)

$$\text{softmax}((\boldsymbol{z}_{n(\cdot,j)} + \boldsymbol{\delta}_j^{(t)})^\top(\boldsymbol{z}_{n(\cdot,k)} + \boldsymbol{\delta}_k^{(t)})) = \Theta(\frac{1}{P})$$
(A47)

$$\text{softmax}((\boldsymbol{z}_{n(\cdot,j)} + \boldsymbol{\delta}_j^{(t)})^\top(\boldsymbol{z}_{n(\cdot,j)} + \boldsymbol{\delta}_j^{(t)})) \geq \frac{1}{P}$$
(A48)

$$\text{softmax}((\boldsymbol{z}_{n(\cdot,k)} + \boldsymbol{\delta}_k^{(t)})^\top(\boldsymbol{z}_{n(\cdot,j)} + \boldsymbol{\delta}_j^{(t)})) = \Theta(\frac{1}{P})$$
(A49)

Therefore, when $k\mu > \Theta(1)$, i.e., $t \geq t_0 = 4P\eta^{-1}(1 - \sigma P)^{-1}$ we have

$$\begin{aligned}
\boldsymbol{\delta}_j^{(t)} &= \mathbb{E}[\boldsymbol{\delta}_j^{(t)}] + \frac{\eta t}{P}\sqrt{\frac{\log B(t - t_0)}{B(t - t_0)}}(\pm(\boldsymbol{e}_1 + \boldsymbol{e}_2) \pm \frac{1}{P}e^{-P^4}(\boldsymbol{e}_3 + \boldsymbol{e}_4)) \\
&= \mathbb{E}[\boldsymbol{\delta}_j^{(t_0)}] - \mathbb{E}[\eta\sum_{b=t_0}^{t}\frac{1}{B}\sum_{n\in\mathcal{B}_b}\frac{\partial}{\partial\boldsymbol{\delta}_j}\ell(f_{\boldsymbol{\theta}}(\boldsymbol{x}_n, \boldsymbol{\delta}^{(b)}), y_n)] \pm \frac{\eta t}{P}\sqrt{\frac{\log B(t - t_0)}{B(t - t_0)}}(\boldsymbol{e}_1 + \boldsymbol{e}_2) \\
&= \mathbb{E}[\boldsymbol{\delta}_j^{(t_0)}] + \frac{\eta(t - t_0)}{4P}(\lambda\boldsymbol{e}_1 + \lambda\boldsymbol{e}_2 + \mu\boldsymbol{e}_3 + \mu\boldsymbol{e}_4) \pm \frac{\eta t}{P}\sqrt{\frac{\log B(t - t_0)}{B(t - t_0)}}(\boldsymbol{e}_1 + \boldsymbol{e}_2),
\end{aligned}$$
(A50)

where $\lambda \gtrsim (1 - \sigma)^{-1}$. Then,

$$\left|\boldsymbol{e}_3^\top\mathbb{E}[\eta\sum_{b=t_0}^{t}\frac{1}{B}\sum_{n\in\mathcal{B}_b}\frac{\partial}{\partial\boldsymbol{\delta}}\ell(f_{\boldsymbol{\theta}}(\boldsymbol{x}_n, \boldsymbol{\delta}^{(b)}), y_n)]\right| \lesssim \eta(t - t_0)\frac{1}{P}\cdot\sqrt{\frac{\log B(t - t_0)}{B(t - t_0)}}$$
(A51)

$$\left|\boldsymbol{e}_4^\top\mathbb{E}[\eta\sum_{b=t_0}^{t}\frac{1}{B}\sum_{n\in\mathcal{B}_b}\frac{\partial}{\partial\boldsymbol{\delta}}\ell(f_{\boldsymbol{\theta}}(\boldsymbol{x}_n, \boldsymbol{\delta}^{(b)}), y_n)]\right| \lesssim \eta(t - t_0)\frac{1}{P}\cdot\sqrt{\frac{\log B(t - t_0)}{B(t - t_0)}}$$
(A52)

and thus $|\mu| \leq \Theta(1/\sqrt{B(t - t_0)})$. Hence, for $\boldsymbol{z}_{n(\cdot,k)}$ from $\boldsymbol{v}_1$ and $\boldsymbol{z}_{n(\cdot,j)}$ from $\boldsymbol{v}_4$,

$$\begin{aligned}
&(\boldsymbol{z}_{n(\cdot,k)} + \boldsymbol{\delta}_k^{(t)})^\top(\boldsymbol{z}_{n(\cdot,k)} + \boldsymbol{\delta}_k^{(t)}) - (\boldsymbol{z}_{n(\cdot,k)} + \boldsymbol{\delta}_k^{(t)})^\top(\boldsymbol{z}_{n(\cdot,j)} + \boldsymbol{\delta}_j^{(t)}) \\
&= \Theta(1)\cdot\frac{e^{\beta_2^2}}{e^{\beta_2^2} + P - 1}(\frac{e^{\beta_2^2}}{e^{\beta_2^2} + P - 1} + \boldsymbol{e}_1^\top\boldsymbol{\delta}^{(t)})
\end{aligned}$$
(A53)

$$\begin{aligned}
&(\boldsymbol{z}_{n(\cdot,j)} + \boldsymbol{\delta}_j^{(t)})^\top(\boldsymbol{z}_{n(\cdot,k)} + \boldsymbol{\delta}_k^{(t)}) - (\boldsymbol{z}_{n(\cdot,j)} + \boldsymbol{\delta}_j^{(t)})^\top(\boldsymbol{z}_{n(\cdot,j)} + \boldsymbol{\delta}_j^{(t)}) \\
&= \Theta(1)\cdot\frac{e^{\beta_2^2}}{e^{\beta_2^2} + P - 1}\cdot\boldsymbol{e}_1^\top\boldsymbol{\delta}^{(t)}
\end{aligned}$$
(A54)

Since that $\beta_2 = \Theta(1)$, we have

$$\text{softmax}((\boldsymbol{z}_{n(\cdot,k)} + \boldsymbol{\delta}_k^{(t)})^\top(\boldsymbol{z}_{n(\cdot,k)} + \boldsymbol{\delta}_k^{(t)})) = \frac{e^{\Theta(1)\cdot\frac{\boldsymbol{e}_1^\top\boldsymbol{\delta}^{(t)}}{P}}}{P - 1 + e^{\Theta(1)\cdot\frac{\boldsymbol{e}_1^\top\boldsymbol{\delta}^{(t)}}{P}}}$$
(A55)

$$\text{softmax}((\boldsymbol{z}_{n(\cdot,k)} + \boldsymbol{\delta}_k^{(t)})^\top(\boldsymbol{z}_{n(\cdot,j)} + \boldsymbol{\delta}_j^{(t)})) = \frac{e^{\Theta(1)\cdot\frac{\boldsymbol{e}_1^\top\boldsymbol{\delta}^{(t)}}{P}}}{P - 1 + e^{\Theta(1)\cdot\frac{\boldsymbol{e}_1^\top\boldsymbol{\delta}^{(t)}}{P}}}$$
(A56)

To make

$$f_{\boldsymbol{\theta}}(\boldsymbol{x}_n, \boldsymbol{\delta}^{(t)}) \geq 1/P,$$
(A57)

we require that

$$\frac{e^{\Theta(1)\cdot\frac{e_1^\top \delta^{(t)}}{P}}}{P-1+e^{\Theta(1)\cdot\frac{e_1^\top \delta^{(t)}}{P}}}\cdot\frac{e^{\beta_2^2}}{e^{\beta_2^2}+P-1}+\frac{P-1}{P-1+e^{\Theta(1)\cdot\frac{e_1^\top \delta^{(t)}}{P}}}\cdot\frac{1}{e^{\beta_2^2}(P-1)+1}\geq\frac{1}{P} \quad \text{(A58)}$$

As a result, we finally need

$$e^{\Theta(1)\cdot\frac{e_1^\top \delta^{(t)}}{P}}\gtrsim P \quad \text{(A59)}$$

which holds as long as $t-t_0\gtrsim P^2\eta^{-1}(1-\sigma)^{-1}\log P$. Therefore, we have

$$f_{\boldsymbol\theta}(\boldsymbol x_n,\boldsymbol\delta)\geq 1/P \quad \text{(A60)}$$

for $\boldsymbol x_n$ that contains a patch from $\boldsymbol v_1$. We similarly have

$$f_{\boldsymbol\theta}(\boldsymbol x_n,\boldsymbol\delta)\leq -1/P \quad \text{(A61)}$$

for $\boldsymbol x_n$ that contains a patch from $\boldsymbol v_2$. To sum up, we need $t\geq\Theta(\eta^{-1}P^2(1-\sigma)^{-1}\log P)$ iterations.

### C.4.2 PROOF OF LEMMA 3

**Proof:**
To avoid multiple superscripts, we use $\boldsymbol\delta$ to denote $\boldsymbol\delta^{(h)}$ since that $h$ is fixed in this proof. We use $\boldsymbol\delta^{(t)}$ to denote the update of $\boldsymbol\delta$ at $t$-th iteration. For the network

$$f_{\boldsymbol\theta}(\boldsymbol x_n,\boldsymbol\delta)=\sum_{l=1}^P\sum_{i=1}^m a_{l(i)}^\top\mathrm{Relu}(\sum_{j=1}^P \boldsymbol W_{O_{2(i,\cdot)}}\boldsymbol W_{V_2}\boldsymbol z_{n(\cdot,j)}\mathrm{softmax}(\boldsymbol z_{n(\cdot,j)}{}^\top \boldsymbol W_{K_2}^\top \boldsymbol W_{Q_2}\boldsymbol z_{n(\cdot,l)})) \quad \text{(A62)}$$

where

$$\boldsymbol z_{n(\cdot,j)}=\mathrm{Relu}(\sum_{s=1}^P \boldsymbol W_{O_1}\boldsymbol W_{V_1}(\boldsymbol x_{n(\cdot,P_{s,1})}+\boldsymbol\delta_s)\mathrm{softmax}((\boldsymbol x_{n(\cdot,P_{s,1})}+\boldsymbol\delta_s)^\top \boldsymbol W_{K_1}^\top \boldsymbol W_{Q_1}(\boldsymbol x_j^n+\boldsymbol\delta_j))), \quad \text{(A63)}$$

we have

$$\frac{\partial f_{\boldsymbol\theta}(\boldsymbol x_n,\boldsymbol\delta)}{\partial \boldsymbol\delta_s}=\sum_{j=1}^P\frac{\partial f_{\boldsymbol\theta}(\boldsymbol x_n,\boldsymbol\delta)}{\partial \boldsymbol z_{n(\cdot,j)}}\frac{\partial \boldsymbol z_{n(\cdot,j)}}{\partial \boldsymbol\delta_s} \quad \text{(A64)}$$

Note that $\boldsymbol W_{Q_2}=\beta_2\boldsymbol I$, $\boldsymbol W_{Q_1}=\beta_1\boldsymbol I$, $\boldsymbol W_{K_2}=\beta_2\boldsymbol P_x$, $\boldsymbol W_{K_1}=\beta_1\boldsymbol P_x$,, $\boldsymbol W_{V_2}=\boldsymbol P_x$, $\boldsymbol W_{V_1}=\boldsymbol P_x$, where $\beta_1=\Theta(1)$ and $\beta_2=\Theta(1)$. Therefore,

$$\frac{\partial f_{\boldsymbol\theta}(\boldsymbol x_n,\boldsymbol\delta)}{\partial \boldsymbol z_{n(\cdot,j)}}$$

$$=\sum_{l=1}^P\sum_{i=1}^m \boldsymbol a_{(l)_i}^\top \mathbb{1}[\sum_{s=1}^P \boldsymbol W_{O_{2(i,\cdot)}}\boldsymbol z_{n(\cdot,P_{s,2})}\mathrm{softmax}(\beta_2^2\boldsymbol z_{n(\cdot,P_{s,2})}{}^\top \boldsymbol z_{n(\cdot,l)})]\Big(\mathrm{softmax}(\beta_2^2\boldsymbol z_{n(\cdot,j)}{}^\top \boldsymbol z_{n(\cdot,l)})$$

$$\cdot\boldsymbol W_{O_{2(i,\cdot)}}+\mathbb{1}[j\neq l]\boldsymbol W_{O_{2(i,\cdot)}}\boldsymbol z_{n(\cdot,j)}\cdot \boldsymbol z_{n(\cdot,l)}\cdot(-\mathrm{softmax}(\beta_2^2\boldsymbol z_{n(\cdot,j)}{}^\top \boldsymbol z_{n(\cdot,l)}))$$

$$\cdot\mathrm{softmax}(\beta_2^2\boldsymbol z_{n(\cdot,l)}{}^\top \boldsymbol z_{n(\cdot,l)})+\mathbb{1}[j=l]\boldsymbol W_{O_{2(i,\cdot)}}\boldsymbol z_{n(\cdot,l)}\mathrm{softmax}(\beta_2^2\boldsymbol z_{n(\cdot,l)}{}^\top \boldsymbol z_{n(\cdot,l)})$$

$$\cdot(1-\mathrm{softmax}(\beta_2^2\boldsymbol z_{n(\cdot,l)}{}^\top \boldsymbol z_{n(\cdot,l)}))\boldsymbol z_{n(\cdot,l)}\Big) \quad \text{(A65)}$$

$$\frac{\partial \boldsymbol{z}_{n(\cdot,j)}}{\partial \boldsymbol{\delta}_k}$$

$$= \mathbb{1}[\sum_{s=1}^{P} \boldsymbol{W}_{O_1}(\boldsymbol{x}_{n(\cdot,P_{s,1})} + \boldsymbol{\delta}_s)\text{softmax}((\boldsymbol{x}_{n(\cdot,P_{s,1})} + \boldsymbol{\delta}_s)^\top(\boldsymbol{x}_j^n + \boldsymbol{\delta}_j))]\Big(\text{softmax}((\boldsymbol{x}_j^n + \boldsymbol{\delta}_j)^\top$$

$$\cdot (\boldsymbol{x}_{n(\cdot,l)} + \boldsymbol{\delta}_l))\boldsymbol{W}_{O_1} + \mathbb{1}[k \neq l]\boldsymbol{W}_{O_1}(\boldsymbol{x}_{n(\cdot,k)} + \boldsymbol{\delta}_k) \cdot (\boldsymbol{x}_{n(\cdot,l)} + \boldsymbol{\delta}_l)^\top$$

$$\cdot (-\text{softmax}(\beta_1^2(\boldsymbol{x}_j^n + \boldsymbol{\delta}_j)^\top(\boldsymbol{x}_{n(\cdot,l)} + \boldsymbol{\delta}_l)))\text{softmax}(\beta_1^2(\boldsymbol{x}_{n(\cdot,l)} + \boldsymbol{\delta}_l)^\top(\boldsymbol{x}_{n(\cdot,l)} + \boldsymbol{\delta}_l))$$

$$+ \mathbb{1}[k = l]\boldsymbol{W}_{O_1}(\boldsymbol{x}_{n(\cdot,l)} + \boldsymbol{\delta}_l)(\boldsymbol{x}_{n(\cdot,l)} + \boldsymbol{\delta}_l)^\top$$

$$\cdot \text{softmax}(\beta_1^2(\boldsymbol{x}_{n(\cdot,l)} + \boldsymbol{\delta}_l)^\top(\boldsymbol{x}_{n(\cdot,l)} + \boldsymbol{\delta}_l))$$

$$\cdot (1 - \text{softmax}(\beta_1^2(\boldsymbol{x}_{n(\cdot,l)} + \boldsymbol{\delta}_l)^\top(\boldsymbol{x}_{n(\cdot,l)} + \boldsymbol{\delta}_l))))\Big) \tag{A66}$$

Let $t = 0$. For $y^n = +1$, Note that if $\boldsymbol{x}_n = [\boldsymbol{e}_3, \boldsymbol{e}_3, \cdots, \boldsymbol{e}_3, \boldsymbol{e}_1, \boldsymbol{e}_3, \cdots, \boldsymbol{e}_3]$ without noise, the loss is 0. Hence, we compute the loss from $\boldsymbol{x}_n = [\boldsymbol{e}_4, \boldsymbol{e}_4, \cdots, \boldsymbol{e}_4, \boldsymbol{e}_1, \boldsymbol{e}_4, \cdots, \boldsymbol{e}_4]$.

$$\mathbb{E}[\mathbb{1}[\sum_{s=1}^{P} \boldsymbol{W}_{O_{(i,\cdot)}}(\boldsymbol{x}_{n(\cdot,P_{s,1})} + \boldsymbol{\delta}_{P_{s,1}}^{(t)})\text{softmax}(\beta_1^2(\boldsymbol{x}_{n(\cdot,P_{s,1})} + \boldsymbol{\delta}_{P_{s,1}}^{(t)})^\top(\boldsymbol{x}_{n(\cdot,l)} + \boldsymbol{\delta}_l)) \geq 0]$$

$$= \text{Pr}(\sum_{s=1}^{P} \boldsymbol{W}_{O_{(i,\cdot)}}(\boldsymbol{x}_{n(\cdot,P_{s,1})} + \boldsymbol{\delta}_{P_{s,1}}^{(t)})\text{softmax}(\beta_1^2(\boldsymbol{x}_{n(\cdot,P_{s,1})} + \boldsymbol{\delta}_{P_{s,1}}^{(t)})^\top(\boldsymbol{x}_{n(\cdot,l)} + \boldsymbol{\delta}_l)) \geq 0) \tag{A67}$$

for $\boldsymbol{W}_{O_{(i,\cdot)}} = \boldsymbol{e}_1$ or $\boldsymbol{e}_4$. We can finally show that with a high probability, the above indicator is close to 1. Meanwhile, for $\boldsymbol{W}_{O_{(i,\cdot)}} = \boldsymbol{e}_2$ or $\boldsymbol{e}_3$, the indicator equals 0 or 1 with half probability when $t = 0$. Consider that $\boldsymbol{x}_{n(\cdot,j)}$ comes from $\boldsymbol{v}_4$. In this case, if $\boldsymbol{x}_{n(\cdot,l)}$ comes from $\boldsymbol{v}_1$,

$$\text{softmax}(\beta_1^2(\boldsymbol{x}_{n(\cdot,l)} + \boldsymbol{\delta}_l)^\top(\boldsymbol{x}_{n(\cdot,l)} + \boldsymbol{\delta}_l)) \geq \frac{1}{P} \tag{A68}$$

$$\text{softmax}(\beta_1^2(\boldsymbol{x}_j^n + \boldsymbol{\delta}_j^{(t)})^\top(\boldsymbol{x}_{n(\cdot,l)} + \boldsymbol{\delta}_l)) = \Theta(\frac{1}{P}) \tag{A69}$$

$$\text{softmax}(\beta_2^2 \boldsymbol{z}_{n(\cdot,l)}^\top \boldsymbol{z}_{n(\cdot,l)}) \geq \frac{1}{P} \tag{A70}$$

$$\text{softmax}(\beta_2^2 \boldsymbol{z}_{n(\cdot,j)}^\top \boldsymbol{z}_{n(\cdot,l)}) = \Theta(\frac{1}{P}) \tag{A71}$$

If $\boldsymbol{x}_{n(\cdot,l)}$ comes from $\boldsymbol{v}_4$, then

$$\text{softmax}(\beta_1^2(\boldsymbol{x}_{n(\cdot,l)} + \boldsymbol{\delta}_l^{(t)})^\top(\boldsymbol{x}_{n(\cdot,l)} + \boldsymbol{\delta}_l^{(t)})) \geq \frac{1}{P} \tag{A72}$$

$$\text{softmax}(\beta_1^2(\boldsymbol{x}_j^n + \boldsymbol{\delta}_j^{(t)})^\top(\boldsymbol{x}_{n(\cdot,l)} + \boldsymbol{\delta}_l^{(t)})) = \Theta(\frac{1}{P}) \tag{A73}$$

$$\text{softmax}(\beta_2^2 \boldsymbol{z}_{n(\cdot,l)}^\top \boldsymbol{z}_{n(\cdot,l)}) \geq \frac{1}{P} \tag{A74}$$

$$\text{softmax}(\beta_2^2 \boldsymbol{z}_{n(\cdot,j)}^\top \boldsymbol{z}_{n(\cdot,l)}) = \Theta(\frac{1}{P}) \tag{A75}$$

Then we consider that $\boldsymbol{x}_{n(\cdot,j)}$ comes from $\boldsymbol{v}_1$. In this case, if $\boldsymbol{z}_{n(\cdot,l)}$ comes from $\boldsymbol{v}_1$, then

$$\text{softmax}(\beta_1^2(\boldsymbol{x}_{n(\cdot,j)} + \boldsymbol{\delta}_j^{(t)})^\top(\boldsymbol{x}_{n(\cdot,l)} + \boldsymbol{\delta}_l^{(t)})) \geq \Theta(\frac{1}{P}) \tag{A76}$$

$$\text{softmax}(\beta_2^2 \boldsymbol{z}_{n(\cdot,j)}^\top \boldsymbol{z}_{n(\cdot,l)}) \geq \Theta(\frac{1}{P}) \tag{A77}$$

If $\boldsymbol{x}_{n(\cdot,l)}$ comes from $\boldsymbol{v}_4$,

$$\text{softmax}(\beta_1^2(\boldsymbol{x}_{n(\cdot,j)} + \boldsymbol{\delta}_j^{(t)})^\top(\boldsymbol{x}_{n(\cdot,l)} + \boldsymbol{\delta}_l^{(t)})) = \Theta(\frac{1}{P}) \tag{A78}$$

$$\text{softmax}(\beta_2^2 \boldsymbol{z}_{n(\cdot,j)}{}^\top \boldsymbol{z}_{n(\cdot,l)}) = \Theta(\frac{1}{P}) \tag{A79}$$

Therefore, if $\boldsymbol{x}_{n(\cdot,j)}$ comes from $\boldsymbol{v}_1$,

$$\frac{\partial f_{\boldsymbol{\theta}}(\boldsymbol{x}_n, \boldsymbol{\delta})}{\partial \boldsymbol{\delta}_j^{(t)}} = P \cdot \frac{1}{4P}\lambda(\boldsymbol{e}_1^\top \cdot \frac{1}{P}\boldsymbol{W}_{O_1})^\top = \frac{1}{4P}\boldsymbol{v}_1 + \Theta(\frac{1}{P})(-\boldsymbol{v}_2 + \boldsymbol{v}_3 - \boldsymbol{v}_4), \tag{A80}$$

and if $\boldsymbol{x}_{n(\cdot,j)}$ comes from $\boldsymbol{v}_4$,

$$\frac{\partial f_{\boldsymbol{\theta}}(\boldsymbol{x}_n, \boldsymbol{\delta})}{\partial \boldsymbol{\delta}_j^{(t)}} = -\frac{1}{4P}\mu\boldsymbol{v}_4 + \Theta(\frac{1}{P})(-\boldsymbol{v}_2 + \boldsymbol{v}_3 + \boldsymbol{v}_1), \tag{A81}$$

where $\lambda = \mu = \Theta(1)$. Note that when $t \geq 2$, since the data which contains $\boldsymbol{v}_2$ and $\boldsymbol{v}_3$ would similarly contribute to the overall gradient, there will be a close amount of $\boldsymbol{v}_1$ and $\boldsymbol{v}_2$ in $\boldsymbol{\delta}_s^{(t)}$ and a close amount of $\boldsymbol{v}_3$ and $\boldsymbol{v}_4$ in $\boldsymbol{\delta}_s^{(t)}$. Hence, when $k\mu < \Theta(1)$,

$$\begin{aligned}
\mathbb{E}[\boldsymbol{\delta}_s^{(t)}] &= \mathbb{E}[\boldsymbol{\delta}_s^{(0)}] - \mathbb{E}[\eta \sum_{b=1}^{t} \frac{1}{B} \sum_{n \in \mathcal{B}_b} \frac{\partial}{\partial \boldsymbol{\delta}_s} \ell(f_{\boldsymbol{\theta}}(\boldsymbol{x}_n, \boldsymbol{\delta}_s^{(b)}), y_n)] \\
&= \eta t \frac{1}{4P}(\lambda\boldsymbol{v}_1 + \lambda\boldsymbol{v}_2 - \mu\boldsymbol{v}_3 - \mu\boldsymbol{v}_4) \\
&= k(\lambda\boldsymbol{v}_1 + \lambda\boldsymbol{v}_2 - \mu\boldsymbol{v}_3 - \mu\boldsymbol{v}_4),
\end{aligned} \tag{A82}$$

$$\boldsymbol{\delta}_s^{(t)} = \mathbb{E}[\boldsymbol{\delta}_s^{(t)}] + \frac{\eta t}{P}\sqrt{\frac{\log Bt}{Bt}}(\pm\boldsymbol{v}_1 \pm \boldsymbol{v}_2 \pm \boldsymbol{v}_3 \pm \boldsymbol{v}_4) \tag{A83}$$

where $\lambda \geq \Theta(1) \cdot (1 - \sigma P)$, $\mu \geq \Theta(1) \cdot (1 - \sigma P)$ for $t \geq 2$. The term $(1 - \sigma P)$ comes from that for $\boldsymbol{W}_{O_2(i,\cdot)} = \boldsymbol{v}_1$ or $\boldsymbol{v}_4$,

$$\mathbb{E}[\mathbb{1}[\sum_{s=1}^{P} \boldsymbol{W}_{O_{1(i,\cdot)}}(\boldsymbol{x}_{n(\cdot,P_{s,1})} + \boldsymbol{\delta}_{P_{s,1}}^{(t)})\text{softmax}(\beta_1^2(\boldsymbol{x}_{n(\cdot,P_{s,1})} + \boldsymbol{\delta}_{P_{s,1}}^{(t)})^\top (\boldsymbol{x}_{n(\cdot,l)} + \boldsymbol{\delta}_l^{(t)})) \geq 0]$$
$$\geq 1 - e^{\frac{(Bt)^2}{\sigma^2 P^2}} \geq 1 - \sigma P \tag{A84}$$

given $B \geq \Theta(1)$ by Hoeffding inequality. When $k\mu \geq \frac{\Theta(1)}{1+\gamma}$, we have that for $\boldsymbol{x}_{n(\cdot,j)}$ from $\boldsymbol{v}_4$,

$$\begin{aligned}
&\mathbb{1}[\sum_{s=1}^{P} \boldsymbol{W}_{O_1}(\boldsymbol{x}_{n(\cdot,P_{s,1})} + \boldsymbol{\delta}_s)\text{softmax}(\beta_1^2(\boldsymbol{x}_{n(\cdot,P_{s,1})} + \boldsymbol{\delta}_s)^\top (\boldsymbol{x}_{n(\cdot,j)} + \boldsymbol{\delta}_j^{(t)})) \geq 0] \\
&\geq [1, 1, -k\mu + (1 - k\mu)\gamma + \boldsymbol{v}_3^\top \boldsymbol{a}, -k\mu\gamma + 1 - k\mu + \boldsymbol{v}_4^\top \boldsymbol{a}]^\top \\
&\geq [1, 1, 0, 0]^\top
\end{aligned} \tag{A85}$$

where $\boldsymbol{a} \sim \mathcal{N}(0, \sigma^2 \boldsymbol{I})$ in the first step, and the last step holds with probability at least

$$\Pr(\boldsymbol{v}_4^\top \boldsymbol{a} - k\mu\gamma + 1 - k\mu \leq 0) \leq 1 - \Pr(\boldsymbol{v}_4^\top \boldsymbol{a} \geq \Theta(1)) \leq 1 - e^{\frac{1}{\sigma^2}} \leq 1 - e^{-P^2} \tag{A86}$$

$$\Pr(\boldsymbol{v}_3^\top \boldsymbol{a} - k\mu + (1 - k\mu)\gamma \leq 0) \leq 1 - \Pr(\boldsymbol{v}_3^\top \boldsymbol{a} \geq \Theta(1)) \leq 1 - e^{\frac{1}{\sigma^2}} \leq 1 - e^{-P^2} \tag{A87}$$

Hence, for $\boldsymbol{x}_{n(\cdot,k)}$ from $\boldsymbol{v}_1$ and $\boldsymbol{x}_{n(\cdot,j)}$ from $\boldsymbol{v}_4$,

$$(\boldsymbol{x}_{n(\cdot,k)} + \boldsymbol{\delta}_k^{(t)})^\top (\boldsymbol{x}_{n(\cdot,k)} + \boldsymbol{\delta}_k^{(t)}) - (\boldsymbol{x}_{n(\cdot,k)} + \boldsymbol{\delta}_k^{(t)})^\top (\boldsymbol{x}_{n(\cdot,j)} + \boldsymbol{\delta}_j^{(t)}) = \Theta(1) \cdot (1 + 2(k\mu)^2) \tag{A88}$$

$$(\boldsymbol{x}_{n(\cdot,j)} + \boldsymbol{\delta}_j^{(t)})^\top (\boldsymbol{x}_{n(\cdot,k)} + \boldsymbol{\delta}_k^{(t)}) - (\boldsymbol{x}_{n(\cdot,j)} + \boldsymbol{\delta}_j^{(t)})^\top (\boldsymbol{x}_{n(\cdot,j)} + \boldsymbol{\delta}_j^{(t)}) = \Theta(1) \cdot (2k\mu - 1) \tag{A89}$$

Since that $\beta_1 = \Theta(1)$, we have

$$\text{softmax}(\beta_1^2(\boldsymbol{x}_{n(\cdot,k)} + \boldsymbol{\delta}_k^{(t)})^\top (\boldsymbol{x}_{n(\cdot,k)} + \boldsymbol{\delta}_k^{(t)})) = \frac{e^{\Theta(1) \cdot (k\mu)^2}}{P - 1 + e^{\Theta(1) \cdot (k\mu)^2}} \tag{A90}$$

$$\text{softmax}(\beta_1^2(\boldsymbol{x}_{n(\cdot,k)} + \boldsymbol{\delta}_k^{(t)})^\top (\boldsymbol{x}_{n(\cdot,j)} + \boldsymbol{\delta}_j^{(t)})) = \frac{e^{\Theta(1) \cdot k\mu}}{P - 1 + e^{\Theta(1) \cdot k\mu}} \tag{A91}$$

To make

$$f_{\boldsymbol{\theta}}(\boldsymbol{x}_n, \boldsymbol{\delta}^{(t)}) \geq 1/P, \tag{A92}$$

we require that

$$\frac{e^{\Theta(1) \cdot (k\mu)^2}}{P - 1 + e^{\Theta(1) \cdot (k\mu)^2}} \cdot 1 \geq \frac{1}{P} \tag{A93}$$

or

$$\frac{e^{\Theta(1) \cdot k\mu}}{P - 1 + e^{\Theta(1) \cdot k\mu}} \cdot 1 \geq \frac{1}{P} \tag{A94}$$

As a result, we finally need

$$e^{\Theta(1) \cdot k\mu} \gtrsim 1 \tag{A95}$$

which holds as long as $t \gtrsim P\eta^{-1}(1 - P\sigma)^{-1}(1 + \gamma)^{-1})$. With the same condition, we also have that for all $y^n = -1$,

$$f_{\boldsymbol{\theta}}(\boldsymbol{x}_n, \boldsymbol{\delta}) \leq -1/P \tag{A96}$$

To sum up, we need $t \geq \Theta(P\eta^{-1}(1 - P\sigma)^{-1}(1 + \gamma)^{-1}))$.

