# OpenReview forum: "Visual Prompting Reimagined: The Power of Activation Prompts"
_ICLR.cc/2024/Conference — Submitted to ICLR 2024_

### Official Review · Reviewer_Z13f · 2023-10-30

**Soundness:** 3 good
**Presentation:** 2 fair
**Contribution:** 3 good
**Rating:** 5
**Confidence:** 4

**Summary:**

In this work, the authors propose to perform the prompting to internal layers of the architecture instead of simply at the input. In a nutshell, they learn the perturbation vector (so, an additive prompting is treated) in the internal layers of the model. It is discussed in the paper that such an approach focuses more on deeper layers, which favors computational complexity at downstream task adaptation phases. the authors provide some theoretic boundaries and connection to normalization approaches, and benchmark on some known datasets for prompt approaches and on two architectures- a ViT and a ResNet.

**Strengths:**

- This paper proposes an intuitive extension of the use of prompts.
- The approach has some grounded connections with x-normalization approaches, extensively discussed (although quite intuitive)
- The chosen datasets are pertinent for the specific type of paper

**Weaknesses:**

- The performance reported in Tab.4 shows that AP is not clearly superior to any other approach, but is in between other approaches.
- It is unclear how to choose the layer(s) where to employ AP.
- The presentation of the paper is very dense, and a few concepts at a time are presented. The message of the paper is at its core very simple and the proposed approach, in the reviewer's opinion there is no need to make it look more complicated than it is in reality.
- The evaluation of one ViT architecture only and one ResNet seem insufficient to validate the approach

**Questions:**

-have the authors tested the approach on different transformer architectures, like SWIN? And what about shallower models?
- how are the authors choosing the layers to apply AP to? Are the authors using a validation set for such a selection?
- have the authors tried to apply AP to more layers at the same time?
- are the authors willing to release the code open-source?

---

> ### Author Response · Authors · 2023-11-20
> **Point-to-Point Response for Reviewer Z13f (Part I)**
>
> We appreciate the valuable comments and suggestions from the reviewer. Please see our point-to-point response below.
>
> **Q1.** *(**Concerns of the Performance of AP Compared to Other PEFT Methods**) The performance reported in Tab.4 shows that AP is not clearly superior to any other approach, but is in between other approaches.*
>
> **A1**: We thank the reviewer for raising this question. We acknowledge that AP may not surpass all PEFT methods. However, it's important to underscore that our focus is on deepening the understanding of the (in)effectiveness of the input-level VP and providing empirical explanation plus theoretical proof to further advance it, rather than developing a method that can outperform all PEFT methods in every scenario. We view the reduced gap between AP and SOTA PEFT methods as a significant positive outcome, showing the progress of advancing the input-level VP.
>
> The above rationale underpins our focus on comparing AP primarily with the state-of-the-art VP method in our major experiments (Tables 1, 2, and 3). The comparison with PEFT methods in Table 4 is included to illustrate that AP's effectiveness is on par with other PEFT methods, while also highlighting its efficiency advantages. We refer readers to GR 1 “Clarification on the focus of our research” for more details.
>
>
> **Q2.** *(**How to Choose Layer for AP on Different Models?**) It is unclear how to choose the layer(s) where to employ AP.*
>
> **A2**: Thank you for your question regarding the optimal layer selection for employing AP in different models. As we have elaborated in Section 4 and illustrated in Figure A1 of our manuscript, extensive experiments were conducted to discern layer preferences in different model architectures. These studies reveal a quite robust choice:  For CNN models, applying AP closer to the output layers (deeper layers) is most effective, whereas for ViT models, the earlier layers (shallower layers) yield better results.
>
> In line with these observations, as outlined in the 'Implementation, Training, and Evaluations' section on Page 7, we suggest employing AP at the input of the third-to-last ResNet block for ResNet models, and at the third Transformer block for ViT models. This recommendation stems from our comprehensive experimental findings and practical experience with these models.
>
> Moreover, our additional experiments featuring CLIP (as discussed in the original submission) and Swin-Transformers (incorporated during the rebuttal phase), presented in [Figure A2](https://imgur.com/a/noH9CKC), validated the scalability of this strategy. These results underscore that the layer selection approach we advocate can be effectively adapted to other model architectures, showing the AP’s practical utility.
>
> **Q3**. *(**The Presentation Looks Complicated**) The presentation of the paper is very dense, and a few concepts at a time are presented. The message of the paper is at its core very simple and the proposed approach, in the reviewer's opinion there is no need to make it look more complicated than it is in reality.*
>
> **A3**: Thanks for the reviewer’s feedback regarding the complexity of our paper's presentation. We recognize that Sections 3 and 4, which delve into the empirical and theoretical core of our work, may appear dense due to the coverage of necessary concepts and mathematical proofs. However, this level of detail is intended to ensure the manuscript's clarity and completeness.
>
> Specifically, Section 3 first reveals the suboptimal design of the current input-level VP by empirically examining AP applied across different layers. This is followed by a theoretical exploration connecting AP with normalization tuning (NT), thereby elucidating AP’s operational mechanism. In Section 4, we investigate AP's optimal configuration for different vision models (ViTs vs. CNNs), uncovering layer preferences and their relation to model architecture. This section also includes empirical analysis linking layer preferences to the models' ability to capture global features, culminating in a formal theoretical framework to support these findings. While we acknowledge that the technical sections of our paper may appear densely packed with information, we have made every effort to include motivating examples, maintain clear explanations, and provide in-depth insights before delving into theoretical results. Therefore, we are confident that our paper is presented in a manner that ensures its clarity and comprehensibility.
>
> Nonetheless, we are open to specific suggestions on how to streamline these sections without compromising their informative value and look forward to further guidance from the reviewer on this matter.

---

> ### Author Response · Authors · 2023-11-20
> **Point-to-Point Response for Reviewer Z13f (Part II)**
>
> **Q4.** *(**More Model Architectures to Test**) The evaluation of one ViT architecture only and one ResNet seem insufficient to validate the approach. Have the authors tested the approach on different transformer architectures, like SWIN? And what about shallower models?*
>
> **A4**: Thanks for your suggestion.
>
> First, in our initial submission, we had already presented results for another vision model CLIP in [Table A2](https://imgur.com/a/noH9CKC) and [Figure A2](https://imgur.com/a/noH9CKC).
>
> Second, in recognition of your suggestion to broaden our model diversity, we conducted further experiments using the Swin-Transformer. The outcomes of these additional studies are detailed in revised [Figure A2](https://imgur.com/a/noH9CKC) and [Table A2](https://imgur.com/a/noH9CKC) in our revised manuscript. The results from these experiments underscore the efficacy of AP as well as the validity of the layer preference proposed in Section 3. Remarkably, AP maintains its enhanced performance over VP across varying architectural complexities, including Swin-Transformer.
>
> Regarding the performance of AP with shallower models, we had conducted experiments and observed that for compact models like ResNet-18 and ViT-Tiny, AP does not outperform NT (Norm-Tune) (although it outpeforms VP). We disclosed this in the “Discussion and Limitations” paragraph on page 9 of our submission. This could be attributed to the fact that AP primarily leverages downstream data to "steer" the existing knowledge acquired during pretraining, rather than actively acquiring new knowledge. As a result, AP tends to favor larger models over more compact ones.
>
>
>
>
> **Q5.** *(**AP on Multiple Layers**) have the authors tried to apply AP to more layers at the same time?*
>
> **A5**: This is an insightful question! During rebuttal, we tried AP with multiple layers and we show the results in [Table A9](https://imgur.com/a/xxvL0zu) in the revision. Our findings indicate that the layer addition of AP does not yield signicant improvements in performance. This observation suggests that applying AP to a single, carefully selected layer can achieve comparable performance to more involved cases.
>
> **Q6.** *(**Source Code**) Are the authors willing to release the code open-source?*
>
> **A6**: Thank you. We have already submitted the source code in the supplementary material along with the paper during the submission. We also refer to the “reproducibility statement” for more details.

---

> ### Author Response · Authors · 2023-11-22
> **Reminder on follow-up discussion (1 day left before rebuttal ends)**
>
> Dear Reviewer Z13f,
>
> We extend our heartfelt appreciation for your dedicated review of our paper. Your efforts are deeply valued by us.
>
> With only 1 day remaining before the conclusion of the discussion phase, we wish to extend a respectful request for your feedback about our general response and point-to-point responses. Your insights are of immense importance to us, and we eagerly anticipate your updated evaluation. Should you find our responses informative and useful, we would be grateful for your acknowledgment. If you have any further inquiries or require additional clarifications, please don't hesitate to reach out. We are fully committed to providing additional responses during this crucial discussion phase.
>
> Best regards,
>
> Authors

---

> > ### Comment · Reviewer_Z13f · 2023-11-22
> > **Response to the rebuttal**
> >
> > I would like to thank the author(s) for their long response. I will address the discussion points in order.
> >
> > 1. As the authors acknowledge, the proposed method is not superior to others in the literature. The authors defend themselves by providing a rationale behind their approach, but this is already accounted for in the paper's strengths.
> >
> > 2. Although the authors propose a quantitative study, I would have expected more theoretical insights on *why* AP needs to be applied in different spots depending on the architecture. "Robustness" claimed in the author's response can be easily challenged by finding empirical counter-examples. What the authors can claim in this perspective is that, for the given examined scenarios, the quantitative studies show that AP should be applied in specific locations. It is not felt, though, that this is final.
> >
> > 3. I acknowledge the positive attitude of the authors to improve the paper's readability. One suggestion could be to provide at the beginning of the sections a summary of the content, to give hints on the logical flow.
> >
> > 4. Thank you, I am satisfied with this response.
> >
> > 5. This finding is opening the door to a whole set of questions, related to reasons why performance is not improving adding more layers. Such a study (both with theoretical grounding and experiments) is felt necessary, as intuitively the performance should increase - the reason it does not motivate the choice of one layer.
> >
> > 6. Thank you, I assume the answer to the question would be "Yes".

---

> > > ### Author Response · Authors · 2023-11-22
> > > **Further Response to Reviewer Z13f**
> > >
> > > Dear Reviewer Z13f,
> > >
> > > We are grateful for your time and efforts on helping us improve the quality of this paper and we thank you for the further comments and suggestions. For each comment (**C1-C6**), we provide our further responses (**R1-R6**) below.
> > >
> > > ***C1**. As the authors acknowledge, the proposed method is not superior to others in the literature. The authors defend themselves by providing a rationale behind their approach, but this is already accounted for in the paper's strengths.*
> > >
> > > **R1.** We appreciate that the reviewer acknowledged the strengths of this paper. While we acknowledge AP may not be the best PEFT methods in terms of accuracy performance, we believe the currently-achieved performance improvement over conventional VP  has been a big step, especially in VP for CNNs, where the existing SOTA method [1] still cannot outperform LP (linear probe). In the area of ViT, although AP does not outperform other PEFT methods significantly, AP still manages to reduce the performance gap between VP and other methods and achieve the best efficiency.
> > >
> > > > [1] Understanding and improving visual prompting: A label-mapping perspective, Aochuan Chen, Yuguang Yao, Pin-Yu Chen, Yihua Zhang, and Sijia Liu, CVPR 2023.
> > >
> > > ***C2**. Although the authors propose a quantitative study, I would ... It is not felt, though, that this is final.*
> > >
> > > **R2.** We thank the reviewer for raising this concern.
> > >
> > > First, we will provide more insights from our current theoretical analyses. It is worth noting that the “layer preference and architecture effect” revealed in this paper is supported by a critical theoretical proof in Appendix C.4. Our theoretical results show that, to achieve the desired generalization performance, shallow-layer AP tuning requires less sample complexity than deep-layer ones for ViTs, which is also empirically verified in Figure 6.
> > >
> > > Second, per the reviewer’s suggestion, we will revise our presentation more carefully and tune down the “robustness” claim. The key point that we want to emphasize is that the “layer preference and architecture effect” revealed in this paper is more than just a heuristic and empirical finding.
> > >
> > > ***C3**. I acknowledge the positive attitude of the authors to improve the paper's readability. One suggestion could be to provide at the beginning of the sections a summary of the content, to give hints on the logical flow.*
> > >
> > > **R3.** Thanks for the concrete suggestions and we agree it is very helpful to add a summary at the beginning of each technical section. Specifically, in Section 3, we will add the following section overview:
> > >
> > > > In this section, we begin by introducing the mathematical formulation of the conventional input-level VP and then expand this VP concept into a more versatile form that we call AP, serving as a foundational element for gaining deeper insights into VP and operating as a highly effective and flexible prompting technique. Moreover, we show that the traditional VP is neither as effective nor as efficient as AP. Lastly, we connect AP and NormTune to better understand AP's operational mechanism.
> > >
> > > In Section 4, we will add the following section overview:
> > >
> > > > In this section, we aim to address the question: When and under which conditions does AP achieve optimal performance? Building upon the insights gained from our preliminary study in Sec. 3, this section launches a deeper exploration into the layer preferences of AP across different models. Our results unveil significant variations in AP's layer preferences depending on the model architecture type. Furthermore, we offer an empirical explanation to enlighten the underlying causes and supplement it with a rigorous theoretical justification.
> > >
> > > However, due to the page limit, we are not able to reflect this in the revision right away. We will revise the manuscript accordingly when more pages are granted.
> > >
> > > ***C4**. Thank you, I am satisfied with this response.*
> > >
> > > **R4.** We are encouraged to learn that the concern has been addressed. Thanks for the insightful question again.
> > >
> > > ***C5**. This finding is opening the door to a whole set of questions, ... does not motivate the choice of one layer.*
> > >
> > > **R5.** We appreciate that the reviewer raises this concern. We agree that this is an important question and deserves more investigations. In our experiments, we noticed that there is a clear tradeoff between the optimization difficulty and the AP performance when more layers are installed with AP. In order to gain significant performance improvements, optimal hyper-parameters for different layers might need to be tuned separately. This is beyond our ability within the short time window of the rebuttal phase. However, we promise to make a more in-depth study on multi-layer AP, including the hyper-parameter variance and its relation to downstream task difficulty. Thank you for this suggestion again.
> > >
> > > ***C6**. Thank you, I assume the answer to the question would be "Yes".*
> > >
> > > **R6.** Yes, we will open source the entire code repository.

---

### Official Review · Reviewer_TRtL · 2023-11-04

**Soundness:** 4 excellent
**Presentation:** 4 excellent
**Contribution:** 3 good
**Rating:** 6
**Confidence:** 2

**Summary:**

This paper proposed a novel visual prompt strategy, termed activation prompt, to reduce the performance gap between visual prompting and widely adopted fine-tuning. Differing from the visual prompt (adding perturbation to the input images), the activation prompt also adds perturbation to the activation maps in the intermediate layers of the model. From both empirical and theoretical aspects, the authors showed the connection between activation prompts and normalization tuning and also ablated the effect of the layer and architecture. Experiments on 29 datasets showed improved performance of activation prompts over the conventional visual prompts and getting closer performance to fine-tuning methods.

**Strengths:**

+ The manuscript is composed with clarity, presenting a concept that is both coherent and well-motivated.
+ The experimental setting is clearly stated, and the authors have conducted comparisons with a wide array of baseline methods.
+ The scope of experimentation is thorough. Despite activation prompting not yet reaching the performance of complete fine-tuning, it notably surpasses traditional visual prompting and a range of other methods aiming for efficient fine-tuning.

**Weaknesses:**

No major weakness is detected. Please refer to questions and suggestions in the following section.

Minor concern:

- It is surprising that parameter-efficient fine-tuning has such a large performance gap compared with full fine-tuning. This is inconsistent with the literature, such as Hu et al., where LoRA achieved similar or even higher performance than full fine-tuning. LoRA served as one of the baseline methods in Table 4, but its performance is much lower than full fine-tuning.

**Reference**

Hu, Edward J., Yelong Shen, Phillip Wallis, Zeyuan Allen-Zhu, Yuanzhi Li, Shean Wang, Lu Wang, and Weizhu Chen. "Lora: Low-rank adaptation of large language models." arXiv preprint arXiv:2106.09685 (2021).

**Questions:**

1. It is unclear why the authors compare activation prompting with *normalization tuning*. Although the related work section includes a citation for normalization tuning, there should be a more comprehensive introduction to this concept and its relevance to activation prompting in the introduction. It would be beneficial for the authors to clarify why normalization tuning was selected as a primary baseline for comparison in Tables 1 and 2.

2. There appears to be a notable performance disparity between the proposed activation prompting method and full fine-tuning. In academia, such a gap may be acceptable if innovative algorithms can bridge it. However, many practical applications in vision prioritize performance over computational efficiency. Could the authors discuss any advantages of prompting-based approaches beyond computational savings? For instance, are these approaches more effective in scenarios requiring few-shot transfer learning, where the data available is much less than that shown in Table 3?

---

> ### Author Response · Authors · 2023-11-20
> **Point-to-Point Response for Reviewer TRtL (Part I)**
>
> We extend our sincere gratitude for your insightful and detailed comments on our manuscript. We particularly appreciate your comprehensive summary of our contributions, including both our empirical and theoretical findings. Please see a point-to-point response below.
>
> **Q1** *(**The Concerns on the performance of LoRA**) It is surprising that parameter-efficient fine-tuning has such a large performance gap compared with full fine-tuning. This is inconsistent with the literature, such as Hu et al., where LoRA achieved similar or even higher performance than full fine-tuning. LoRA served as one of the baseline methods in Table 4, but its performance is much lower than full fine-tuning.*
>
> **A1.** We acknowledge the reviewer's concern regarding the performance disparity between LoRA and full fine-tuning as observed in our study. This variation primarily stems from the different types of tasks being evaluated. Specifically, the LoRA paper [1] focuses on NLP tasks, which markedly differ from the visual classification tasks addressed in our research. Consequently, this distinction may account for the varying performance levels of LoRA observed in our study compared to [1]. To further alleviate the reviewer’s concern, we conduct additional experiments on the hyper-parameters of LoRA, namely the rank r. In our original submission, the rank $r$ is adopted to 10 by default. In [Table A8](https://imgur.com/a/r2AQpdm) of our revised manuscript, we explore more rank values varying from 1 to 50. We can see that the performance of LoRA increases with the larger rank values, but the accuracy difference between r=10 and r=50 is not significant. In contrast, the efficiency of LoRA will drop significantly with a rank larger than 10. In order to strike a balance between accuracy and efficiency, we adopt the rank value of 10 as the default value in this work.
>
> > [1] LoRA: Low-Rank Adaptation of Large Language Models, Edward J Hu, Yelong Shen, Phillip Wallis, Zeyuan Allen-Zhu, Yuanzhi Li, Shean Wang, Lu Wang, and Weizhu Chen, ICLR 2022.
>
>
> **Q2** *(**The Connection between AP and Normalization Tuning**) It is unclear why the authors compare activation prompting with normalization tuning. Although the related work section includes a citation for normalization tuning, there should be a more comprehensive introduction to this concept and its relevance to activation prompting in the introduction. It would be beneficial for the authors to clarify why normalization tuning was selected as a primary baseline for comparison in Tables 1 and 2.*
>
>
> **A2.** We appreciate the reviewer's question and apologize for any confusion caused. The decision to use Normalization Tuning (NT) as a primary baseline in our experiments is rooted in its conceptual connection to Activation Prompting (AP). As elaborated in the last paragraph of Section 3, we have identified an intrinsic link between AP and a special form of NT, revealing that under certain conditions, these methods can be transformed into each other. This interconnection is crucial for illustrating the functioning mechanism of AP and providing a new perspective on prompting methodologies. To this end, we extend a formal proof on the relationship between these two methods. Additionally, taking the reviewer’s suggestion into account, we have revised the introduction section of our manuscript to incorporate a more comprehensive discussion on NT and its relevance to AP.

---

> ### Author Response · Authors · 2023-11-20
> **Point-to-Point Response for Reviewer TRtL (Part II)**
>
> **Q3** *(**Concern on the Performance Gap Between AP and PEFT Methods**) There appears to be a notable performance disparity between the proposed activation prompting method and full fine-tuning. In academia, such a gap may be acceptable if innovative algorithms can bridge it. However, many practical applications in vision prioritize performance over computational efficiency. Could the authors discuss any advantages of prompting-based approaches beyond computational savings? For instance, are these approaches more effective in scenarios requiring few-shot transfer learning, where the data available is much less than that shown in Table 3?*
>
> **A3.** We thank the reviewer for raising this concern. We acknowledge that the performance of AP can not beat all the PEFT baselines in every aspect. However, as we elaborated in GR1 “Clarification on the focus of our research”, this work aims to leverage AP as a crucial tool to gain theoretical and practical insights of the input-level VP, and further to explore the possible enhancements. Our work has confirmed that the significant performance drop caused by the conventional VP can be alleviated using AP. In addition, there exist other benefits brought by AP.
>
> First (**Multiple Efficiency Metrics**), beyond computational efficiency, AP excels in other efficiency aspects, such as parameter and inference efficiency, as demonstrated in Tables 2 and 4. These advantages position AP as a cost-effective tool post-training, offering benefits in terms of storage and deployment costs.
>
> Second (**Useful Tool for Understanding Model Behaviors**), AP also serves as a valuable tool for probing into the nature of different tyles of model architectures. For instance, in this work, we revealed the distinct layer preference between CNNs and ViTs, as discussed in Section 4.1, and provide insights into how different layers of a model capture global features advantageous to prompts. These analyses can be broadly applied to various model architectures, as backed up by our additional experiments showed in [Figure A2](https://imgur.com/a/noH9CKC) for CLIP and Swin-Transformer models. This understanding also enhances the explainability of model behaviors.
>
> Finally, regarding the reviewer's suggestion of exploring AP in extreme few-shot learning scenarios, our experiments did not show substantial improvements over existing SOTA PEFT methods in such settings. Moreover, as illustrated in GR1 "clarification on the focus of our research", our primary objective is **not** to propose a method that can outperform all PEFT methods in a given scenario. Instead, we view the reduced gap between AP and SOTA PEFT methods as a significant positive outcome, showing the progress of advancing the input-level VP.

---

> ### Comment · Reviewer_TRtL · 2023-11-22
>
> Thank you for the well-prepared response and revision. They have addressed my earlier concerns. But I suggest the authors strengthen the fairness in the comparison (as raised by multiple reviewers) and ensure a sufficient comparison with established baseline methods. I must admit my limited expertise in this particular area (shown in my confidence), so I may not be completely up-to-date with the most recent literature. While the paper's overall presentation is satisfactory, a more equitable and thorough comparison with existing methods is critical for strengthening the paper's contribution to the field.

---

> > ### Author Response · Authors · 2023-11-23
> > **Further Response to Reviewer TRtL**
> >
> > Dear Reviewer TRtL,
> >
> > Thank you for your response, and we are pleased to see that your previous responses have been addressed. In response to the additional suggestion from the reviewer, we would like to emphasize that we have put forth significant effort in our initial submission to ensure a fair and comprehensive evaluation. Building upon the reviewers’ suggestions, we further enhanced our evaluation. These endeavors have been meticulously integrated into our responses to all the reviewers for this paper, as well as our [General Response 2](https://openreview.net/forum?id=0b328CMwn1&noteId=PoENtxqXLp) (GR2).
> >
> > For your convenience, we make a short summary below.
> >
> > * To ensure the direct comparability with VPT, we conducted additional experiments directly on the setting in the VPT paper ([Table A3](https://imgur.com/a/kUJL0tI)), and performed a layerwise comparison with VPT ([Figure A3](https://imgur.com/a/6eQt5Ls)), **to ensure the validity of our experiment results and its fairness;**
> >
> > * Similarly, we compared AP to more variants of the baselines, i.e., VPT with different prompt lengths ([Table A5](https://imgur.com/a/yCg4Nxm)) and LoRA with different ranks ([Table A8](https://imgur.com/a/r2AQpdm));
> >
> > * We compared more PEFT baselines ([Table A7](https://imgur.com/a/nBuTWio)) and finally compared to 5 SOTA PEFT baselines, **to ensure the comprehensiveness of our experiments**;
> >
> > * We conducted more experiments on more model architectures ([Table A2](https://imgur.com/a/GlWCTeH) and [Figure A2](https://imgur.com/a/noH9CKC)) **to ensure the generality of the conclusion of this work**.
> >
> > In the end, we would like to thank you for your invaluable suggestions and contributions to improving the quality of our manuscript again. If you have any other questions or comments, we are happy and open to more discussions.
> >
> > Authors.

---

### Official Review · Reviewer_jt12 · 2023-11-06

**Soundness:** 3 good
**Presentation:** 3 good
**Contribution:** 3 good
**Rating:** 5
**Confidence:** 4

**Summary:**

This paper introduces a novel method called Activation Prompt (AP) as an extension of VPT-deep. AP not only provides a theoretical framework that emphasizes the relationship between AP and normalization tuning but also presents extensive experimental results across 29 datasets from diverse learning benchmarks. These experiments demonstrate a consistent improvement over VP in various learning scenarios.

**Strengths:**

1. This paper proposes a new method, namely activation prompt (AP), which is a modification of VPT-deep.
2. Authors conduct extensive experimentation involving 29 datasets across various benchmarks.

**Weaknesses:**

1. This paper presents a close relationship with VPT-Deep [1]. The authors propose a modification by replacing the concatenation operation in VPT-Deep with a summation operation. While this modification is relatively moderate, the authors claim that AP exhibits distinct attributes compared to VP, such as reduced computational demands when embedded deeper within the architecture.

2. VPT-Deep could also be applied exclusively to deeper layers. However, in Figure 2, it appears that the authors did not implement a layer-wise comparison between VPT-Deep and AP, which could have provided a fairer assessment.

3. The authors fail to provide a thorough explanation for why AP outperforms VPT-Deep. It seems that the feature-wise prompt is superior to the concatenation prompt. Could alternative strategies like feature-wise product or other feature-wise aggregation methods be considered?

4. Furthermore, while the authors introduce several related PEFT algorithms, they are absent from the experimental section. Notably, algorithms such as LORA [2], SSF [3], and FACT [4] are advanced PEFT approaches that could have been compared against. These algorithms are specifically designed for PEFT and report their performance on the same datasets. I suggest considering including a comparison with these approaches.

[1] Jia M, Tang L, Chen B C, et al. Visual prompt tuning, ECCV, 2022: 709-727.
[2] Hu E J, Shen Y, Wallis P, et al. Lora: Low-rank adaptation of large language models[J]. arXiv preprint arXiv:2106.09685, 2021.
[3] Lian DZhou D, Feng J, et al. Scaling & shifting your features: A new baseline for efficient model tuning, NeurIPS, 2022, 35: 109-123.
[4] Jie S, Deng Z H. Fact: Factor-tuning for lightweight adaptation on vision transformer, AAAI. 2023, 37(1): 1060-1068.

**Questions:**

1. VPT-Deep could also be applied exclusively to deeper layers. However, in Figure 2, it appears that the authors did not implement a layer-wise comparison between VPT-Deep and AP, which could have provided a fairer assessment.

2. The authors fail to provide a thorough explanation for why AP outperforms VPT-Deep. It seems that the feature-wise prompt is superior to the concatenation prompt. Could alternative strategies like feature-wise product or other feature-wise aggregation methods be considered?

3. Furthermore, while the authors introduce several related PEFT algorithms, they are absent from the experimental section. Notably, algorithms such as LORA [2], SSF [3], and FACT [4] are advanced PEFT approaches that could have been compared against. These algorithms are specifically designed for PEFT and report their performance on the same datasets. I suggest considering including a comparison with these approaches.

[1] Jia M, Tang L, Chen B C, et al. Visual prompt tuning, ECCV, 2022: 709-727.
[2] Hu E J, Shen Y, Wallis P, et al. Lora: Low-rank adaptation of large language models[J]. arXiv preprint arXiv:2106.09685, 2021.
[3] Lian DZhou D, Feng J, et al. Scaling & shifting your features: A new baseline for efficient model tuning, NeurIPS, 2022, 35: 109-123.
[4] Jie S, Deng Z H. Fact: Factor-tuning for lightweight adaptation on vision transformer, AAAI. 2023, 37(1): 1060-1068.

---

> ### Author Response · Authors · 2023-11-20
> **Point-to-Point Response for Reviewer jt12 (Part I)**
>
> We thank the reviewer for the constructive comments and suggestions, please see our response below.
>
> **Q1**: *(**Concerns on the Novelty of this Work**) This paper presents a close relationship with VPT-Deep [1]. The authors propose a modification by replacing the concatenation operation in VPT-Deep with a summation operation. While this modification is relatively moderate, the authors claim that AP exhibits distinct attributes compared to VP, such as reduced computational demands when embedded deeper within the architecture.*
>
> A1: Thanks for raising this question and we believe this is a misunderstanding towards the motivation, research focus as well as technical contributions of our work. We kindly direct the reviewer to our GR1 for more explanations. We provide a summary of why our work is fundamentally distinct from the VPT work.
>
> Firstly, as elaborated in [GR1 “**Clarification on the focus of our research**”](https://openreview.net/forum?id=0b328CMwn1&noteId=iN7H5ZJS1i), AP is initially conceived as an extension of traditional "input-level" VP. It serves as a crucial tool to gain theoretical and practical insights, aimed at pinpointing the inherent limitations in standard (input-level) VP and exploring possible enhancements.
>
> Secondly, detailed in [GR1 “**Differences from VPT**”](https://openreview.net/forum?id=0b328CMwn1&noteId=X1clKbIGyG), AP is distinguished from VPT-Deep in three critical areas: the technical difference of the layer prompting configuration, a comprehensive analysis of layer preference and the architectural impacts, and a novel theoretical framework evaluating the (in)effectiveness of traditional VP versus the efficacy of AP.
>
> Thirdly, as outlined in [GR1 “**Contributions of this work**"](https://openreview.net/forum?id=0b328CMwn1&noteId=X1clKbIGyG), it is evident that our research encompasses a unique scope, leading to distinct technical contributions when compared to VPT.
>
> We hope with the clarification, the reviewer’s concern can be alleviated and the reviewer can re-evaluate the novelty of our work.
>
> > [1] Visual prompt tuning, Menglin Jia, Luming Tang, Bor-Chun Chen, Claire Cardie, Serge Belongie, Bharath Hariharan, and Ser-Nam Lim, ECCV 2022.
>
> **Q2**: *(**A Layer-wise Comparison to VPT-Deep**) VPT-Deep could also be applied exclusively to deeper layers. However, in Figure 2, it appears that the authors did not implement a layer-wise comparison between VPT-Deep and AP, which could have provided a fairer assessment.*
>
> **A2**: Thanks for raising this question. Yes, VPT-Deep could also be applied to deeper layers. Yet, VPT-deep integrates prompts starting from one deep layer and extends them to all subsequent layers. This is different from AP, which focuses on applying prompts to a specific single layer; see more explanations in GR1 “**Differences from VPT**”.
>
> Moreover, we are grateful for the suggestion to conduct a layer-wise comparison with VPT-Deep. In line with this, we have conducted an additional experiment for a more detailed layer-wise evaluation; see [Figure A3](https://imgur.com/a/6eQt5Ls) in our revised manuscript. These additional results highlight a consistent layer-architecture influence on VPT-Deep, akin to what we initially observed in our original AP design. This outcome is not unexpected, considering that the implementation of VPT-Deep essentially converges with that of AP when a specific network layer is selected for prompting. A minor divergence lies in the prompt design: VPT-Deep uses concatenation, whereas AP opts for addition when integrating prompts into a network layer.

---

> ### Author Response · Authors · 2023-11-20
> **Point-to-Point Response for Reviewer jt12 (Part II)**
>
> **Q3**: *(**Why does AP Outperform VPT? Other Possible Prompt Type?**) The authors fail to provide a thorough explanation for why AP outperforms VPT-Deep. It seems that the feature-wise prompt is superior to the concatenation prompt. Could alternative strategies like feature-wise product or other feature-wise aggregation methods be considered?*
>
> **A3**: Thanks for raising this question.
>
> Firstly, it is important to clarify that we did not claim the superiority of AP over VPT-Deep in all scenarios. This can be justified from Table 4: VPT slightly outperforms AP in the FGVC tasks. As outlined in GR-1 (Clarification on the focus of our research), the primary motivation and the focus of this paper are to enhance understanding and the performance of the conventional input-level VP. The results presented in Table 4 aim to illustrate that AP indeed narrows the performance gap between VP and other PEFT methods, rather than positioning AP as a consistently better method than VPT. We will make our point clearer.
>
> Additionally, we acknowledge the reviewer’s insightful suggestion more prompt types. In response, we have conducted additional experiments using the feature-wise product and the concatenation prompts, with the findings presented in [Table A6](https://imgur.com/a/N7W1mtT) of our revised manuscript. We observed that our originally proposed AP outperforms its new prompt variants studied in [Table A6](https://imgur.com/a/N7W1mtT) (AP-Product and AP-Concate). We speculate that the advantage of the originally proposed AP may stem from its intrinsic connection to normalization tuning, as discussed in the concluding part of Section 3.
>
> **Q4**: *(**Inclusion of Additional PEFT Algorithms in Experiments**) Furthermore, while the authors introduce several related PEFT algorithms, they are absent from the experimental section. Notably, algorithms such as LORA [2], SSF [3], and FACT [4] are advanced PEFT approaches that could have been compared against. These algorithms are specifically designed for PEFT and report their performance on the same datasets. I suggest considering including a comparison with these approaches.*
>
> **A4**: We appreciate the suggestion to include more advanced PEFT methods in our comparisons. We would like to point out that we have already used LoRA as one PEFT baseline together with other 4 state-of-the-art PEFT baselines in Table 4.
>
> Following the reviewer’s suggestion, we conduct new results and report the results of SSF in [Table A7](https://imgur.com/a/nBuTWio) in the revised manuscript. In particular, we can see SSF is also a competitive method among all the baselines, but is still under AdapterFormer. Compared to AP, SSF yields better performance for the FGVC benchmark but leads to slightly worse accuracy for the VTAB benchmark. In general, SSF ranks approximately the second or the third place among all the PEFT methods. Unfortunately, due to time constraints, we were unable to incorporate FACT into this round of experiments. However, we plan to include it in our upcoming experiments.
>
> Furthermore, we wish to reemphasize that our primary objective is not to propose a method that can outperform all PEFT methods in every scenario. Instead, our focus is on deepening the understanding of the (in)effectiveness of the input-level VP and providing empirical explanation plus theoretical proof to further advance it. We view the reduced gap between AP and SOTA PEFT methods as a significant positive outcome, showing the progress of advancing the input-level VP.

---

> ### Author Response · Authors · 2023-11-22
> **Reminder on follow-up discussion (1 day left before rebuttal ends)**
>
> Dear Reviewer jt12,
>
> We extend our heartfelt appreciation for your dedicated review of our paper. Your efforts are deeply valued by us.
>
> With only 1 day remaining before the conclusion of the discussion phase, we wish to extend a respectful request for your feedback about our general response and point-to-point responses. Your insights are of immense importance to us, and we eagerly anticipate your updated evaluation. Should you find our responses informative and useful, we would be grateful for your acknowledgment. If you have any further inquiries or require additional clarifications, please don't hesitate to reach out. We are fully committed to providing additional responses during this crucial discussion phase.
>
> Best regards,
>
> Authors

---

### Official Review · Reviewer_Yfja · 2023-11-06

**Soundness:** 2 fair
**Presentation:** 3 good
**Contribution:** 2 fair
**Rating:** 5
**Confidence:** 4

**Summary:**

This paper proposes a variant of visual prompting by tuning model activations. The effect of applying the technique to different layers is studied. Experiments on various downstream datasets shows parameter-accuracy advantage over baseline VP and parameter efficient fine-tuning, etc.

**Strengths:**

1. The study of applying AP to different layers provides interest insights related to model interworks such as attention distance.
2. Experimental results are presented with many datasets, settings, and compared with various benchmarks.

**Weaknesses:**

1. Novelty given extensive studies in the VPT paper. The VPT paper studies various different ways of applying the prompt. For example, in Figure 5, they studied the option of adding tokens instead of prepending tokens and observed slightly worse results, which is almost verified in Table 4 of this paper. In Fig. 14 of VPT, they also studied adding prompts in different depths of the network. These previous work almost makes this paper a special case in the search space covered by VPT.
2. Fair comparison with existing works such as VPT. The main results are reported all in different settings (e.g. different backbone, patch size, pretraining, etc.) that prevent direct comparison with numbers reported in the VPT paper, although this work outperforms basic baselines such as VP and norm-tune.
3. Soundness of the parameter counts in tables. For example, in Table 2 with ViT-L and CIFAR-10, how does LINEAR-PROBE have 1M parameters instead of 10K (1024 * 10)? In addition, if other methods (e.g. norm-tune and AP) also train a linear layer on top of the features, then this should be made clear as well. The current reported counts are significantly higher than expected (also see table 5 of VPT).
4. Limited gain of efficiency-accuracy trade-off compared with other methods in Table 4 such as LORA etc. The main advantage of AP is the lower cost but other methods are configured to be more expensive than they should be. e.g. this paper experiments with VPT-DEEP with 50 tokens, but (1) VPT also presents VPT-shallow as a more efficient option, (2) VPT by default searches over a set of prompt lengths and in their table 13, a much smaller average token length is needed, e.g. 10 or 1, (3) VPT applied to a deep ViT-L will put VPT in a disadvantage but this could be easily fixed by prompting a subset of layers. As a result, it is not clear where this paper stands in the extensive ablations of VPT.

**Questions:**

As suggested in the weakness section, numbers directly comparable with those reported in VPT paper will facilitate fair comparison. Please see the weakness section for more details.

---

> ### Author Response · Authors · 2023-11-20
> **Point-to-Point Response for Reviewer Yfja (Part I)**
>
> We thank the reviewer for all the constructive comments. Please see our point-to-point response below.
>
> **Q1**: *(**Concerns Regarding Novelty in Light of VPT's Extensive Studies**) Novelty given extensive studies in the VPT paper. The VPT paper studies various different ways of applying the prompt. For example, in Figure 5, they studied the option of adding tokens instead of prepending tokens and observed slightly worse results, which is almost verified in Table 4 of this paper. In Fig. 14 of VPT, they also studied adding prompts in different depths of the network. This previous work almost makes this paper a special case in the search space covered by VPT.*
>
> **A1**: Thanks for raising this question. While both AP (our work) and VPT share a similar design principle of adding prompts to "intermediate" layers or features, they are, in fact, fundamentally different approaches.
>
> First, as explained in [GR1 “**Clarification on the focus of our research**”](https://openreview.net/forum?id=0b328CMwn1&noteId=iN7H5ZJS1i), AP is introduced as a natural extension of the conventional "input-level" VP. And it serves as a valuable tool for acquiring both theoretical and practical insights, with the aim of uncovering the inherent limitations of traditional VP and exploring potential enhancements.
>
> Second, as explained in [GR1 “**Differences from VPT**”](https://openreview.net/forum?id=0b328CMwn1&noteId=X1clKbIGyG), AP differs from VPT (as well as its deep variants) in three key aspects: technical difference in layer prompting configuration, in-depth analysis regarding layer preference and the impact of architecture (CNNs vs. ViTs), a unique theoretical perspective on the (in)effectiveness of traditional VP vs. the effectiveness of AP.
>
> Third, as explained in [GR1 “**Contributions of this work**”](https://openreview.net/forum?id=0b328CMwn1&noteId=X1clKbIGyG), it should be clear that our research has a distinct scope compared to VPT, leading to different technical contributions.
>
> We hope with the clarification, the reviewer’s concern can be alleviated and the reviewer can re-evaluate the novelty of our work.
>
> **Q2**: *(**Concerns About Fair Comparison with VPT**) Fair comparison with existing works such as VPT. The main results are reported all in different settings (e.g. different backbone, patch size, pretraining, etc.) that prevent direct comparison with numbers reported in the VPT paper, although this work outperforms basic baselines such as VP and norm-tune.*
>
> **A2**: Thanks for raising this question. Our experimental setup does not entirely coincide with the VPT framework, primarily because we adopted the testbed used in the SOTA VP method [1] as our principal evaluation framework [1],in line with our core objective detailed in [GR1 “**Contributions of this work**”](https://openreview.net/forum?id=0b328CMwn1&noteId=X1clKbIGyG).” Our aim is to explore the inherent limitations of current VP designs, leveraging AP to comprehensively analyze, understand, prove, and further enhance it. While our experimental setting does not align entirely with the VPT framework, it is not intended to disadvantage VPT. Notably, the differences lie in minor aspects like backbone patch sizes and pretraining methods. However, critical elements such as benchmark datasets remain consistent and are widely recognized in both VP and PEFT fields. Therefore, we firmly believe that our  approach ensures a balanced and fair experimental design.
>
> In the meantime, we also understand the reviewer’s concern. Thus, we conducted additional ablation studies to strictly follow the experiment settings of VPT with these results included in [Table A3](https://imgur.com/a/kUJL0tI) of our revised manuscript. The performance of VPT is directly sourced from [2; Table 1] (VPT paper). As we can see, the performance as well as efficiency of AP positions itself between VPT-Shallow and VPT-Deep, with an average of 3% performance gain over VPT-Shallow and an average of 3.5% drop compared to VPT-Deep. These findings indicate that AP's performance and efficiency are intermediate between VPT-Shallow and VPT-Deep, exhibiting an average performance gain of 3% over VPT-Shallow and a 3.5% decrease compared to VPT-Deep. It is important to note that the VPT results in [2; Table 1] are based on the best prompt length for each dataset, whereas AP employs consistent hyper-parameters across all datasets. We emphasize that our intention in presenting the results in Tables 4 and A3 is not to claim AP's superiority over VPT but to demonstrate that AP successfully narrows the performance gap between the conventional (input-level) VP and leading PEFT methods like VPT.
>
> > [1] Understanding and improving visual prompting: A label-mapping perspective, Aochuan Chen, Yuguang Yao, Pin-Yu Chen, Yihua Zhang, and Sijia Liu, CVPR 2023.
>
> > [2] Visual prompt tuning, Menglin Jia, Luming Tang, Bor-Chun Chen, Claire Cardie, Serge Belongie, Bharath Hariharan, and Ser-Nam Lim, ECCV 2022.

---

> ### Author Response · Authors · 2023-11-20
> **Point-to-Point Response for Reviewer Yfja (Part II)**
>
> **Q3**: *(**Clarifying Parameter Counts in Tables**) Soundness of the parameter counts in tables. For example, in Table 2 with ViT-L and CIFAR-10, how does LINEAR-PROBE have 1M parameters instead of 10K (1024 * 10)? In addition, if other methods (e.g. norm-tune and AP) also train a linear layer on top of the features, then this should be made clear as well. The current reported counts are significantly higher than expected (also see table 5 of VPT).*
>
> **A3**: We apologize for any confusion regarding the parameter count presented in Table 2. The model initially pre-trained on ImageNet, which  includes a classification head comprising 1M parameters (1000 * 1024). For downstream tasks like CIFAR-10, there is an option to either fine-tune this original head or replace it with a re-initialized head of 10K parameters (10 * 1024). Consistent with the approach in the SOTA VP paper [1], we opted for the former in our experiment design. To address the reviewer’s concerns, we carried out additional ablation experiments using the re-initialized classification head. The results, now included in [Table A4](https://imgur.com/a/AUYs6j7) of our revised manuscript, are nearly identical to our previous findings. We also want to remark that the parameter counts for all methods involve the training of the same, consistent linear layer, which thus ensures a fair comparison across different methods. Furthermore, we have taken into consideration the reviewer's suggestion to clarify the usage of the linear probe and have made corresponding revisions to Table 2 and Table 4.
>
> > [1] Understanding and improving visual prompting: A label-mapping perspective, Aochuan Chen, Yuguang Yao, Pin-Yu Chen, Yihua Zhang, and Sijia Liu, CVPR 2023.
>
> **Q4**: *(**Concerns on the VPT Efficiency**) The main advantage of AP is the lower cost but other methods are configured to be more expensive than they should be. e.g. this paper experiments with VPT-DEEP with 50 tokens, but (1) VPT also presents VPT-shallow as a more efficient option, (2) VPT by default searches over a set of prompt lengths and in their table 13, a much smaller average token length is needed, e.g. 10 or 1, (3) VPT applied to a deep ViT-L will put VPT in a disadvantage but this could be easily fixed by prompting a subset of layers. As a result, it is not clear where this paper stands in the extensive ablations of VPT.*
>
> **A4**: We appreciate the opportunity to clarify our choice of prompt length in initial VPT experiments. In our initial experiments with VPT, we adopted a prompt length of 50 tokens for VPT, with the following reasons.
>
> First, there is no official recommendation about the  prompt length in the VPT paper. As shown in  [2, Figure 6], the prompt length of 1 to 200 is tested, and it is concluded that “the optimal prompt length varies across tasks”. Given that our objective is not to position AP as an evolution of VPT, as clarified in [GR1 ("**Clarification on the focus of our research**")](https://openreview.net/forum?id=0b328CMwn1&noteId=iN7H5ZJS1i), we have selected a token length of 50 from the middle of the tested range (1~200). This choice facilitates experimentation and aligns with one of the prominent token lengths highlighted in [2, Figure 6].
>
> Second, although VPT-Shallow is a more efficient solution, as shown in [Table A3](https://imgur.com/a/kUJL0tI), it somehow underperforms VPT-Deep. Thus, in what follows, we conducted an additional experiment to implement VPT-Deep using a smaller prompt token length of 10 (VPT-10). The results, presented in [Table A5](https://imgur.com/a/yCg4Nxm) of our revised manuscript, indicate that VPT-10's performance is comparable to VPT-50. Hence, the results in Table 4 have been updated to include VPT-10 data. We did not choose 1 as the prompt length as it typically compromises the performance of VPT as shown in [2, Figure 6].
>
> Additionally, regarding the comment “As a result, it is not clear where this paper stands in the extensive ablations of VPT”, we would like to reiterate, as detailed in our response A1, that our research is driven by a distinct motivation, encompasses a different research scope (e.g., including VP for both CNNs and ViTs), and offers entirely different technical contributions compared to VPT.
>
> > [2] Visual prompt tuning, Menglin Jia, Luming Tang, Bor-Chun Chen, Claire Cardie, Serge Belongie, Bharath Hariharan, and Ser-Nam Lim, ECCV 2022.

---

> ### Author Response · Authors · 2023-11-22
> **Reminder on follow-up discussion (1 day left before rebuttal ends)**
>
> Dear Reviewer Yfja,
>
> We extend our heartfelt appreciation for your dedicated review of our paper. Your efforts are deeply valued by us.
>
> With only 1 day remaining before the conclusion of the discussion phase, we wish to extend a respectful request for your feedback about our general response and point-to-point responses. Your insights are of immense importance to us, and we eagerly anticipate your updated evaluation. Should you find our responses informative and useful, we would be grateful for your acknowledgment. If you have any further inquiries or require additional clarifications, please don't hesitate to reach out. We are fully committed to providing additional responses during this crucial discussion phase.
>
> Best regards,
>
> Authors

---

### Author Response · Authors · 2023-11-20
**Highlighted General Response (GR) - Part III**

## GR 2: A summary of additional experiments.

The authors have made a substantial effort to enrich our experiments (see the revised PDF). Below is a summary of them, where Q$i$ represents the $i$th question in our response.

* **Reviewer Yfja**
    * **Q2**: Performance comparison following the precise setting of VPT ([Table A3](https://imgur.com/a/kUJL0tI))
    * **Q3**: Experiments with re-initialized classification head ([Table A4](https://imgur.com/a/AUYs6j7))
    * **Q4**: Experiments with VPT with smaller prompt token length ([Table A5](https://imgur.com/a/yCg4Nxm))
* **Reviewer jt12**
    * **Q2**: Layer-wise performance comparison with VPT ([Figure A3](https://imgur.com/a/6eQt5Ls))
    * **Q3**: AP using other prompt implementations ([Table A6](https://imgur.com/a/N7W1mtT))
    * **Q4**: Comparison with more PEFT baselines ([Table A7](https://imgur.com/a/nBuTWio))
* **Reviewer TRtL**
    * **Q1**: Experiments with LoRA at different ranks ([Table A8](https://imgur.com/a/r2AQpdm))
    * **Q3**: Experiments with more model architectures ([Figure A2](https://imgur.com/a/noH9CKC))
* **Reviewer Z13f**
    * **Q2**, **Q4**: Experiments with more model architectures ([Figure A2](https://imgur.com/a/noH9CKC))
    * **Q5**: Experiments with AP installed on multiple layers ([Table A9](https://imgur.com/a/xxvL0zu))

---

### Author Response · Authors · 2023-11-20
**Highlighted General Response (GR) - Part II**

**Differences from VPT.** The study of this work is novel and different from the VPT paper for several reasons.

* (**Technical Difference with VPT**) Strictly speaking, AP and VPT have different designs. AP focuses on applying prompts to a specific model layer. In contrast, VPT with its deep variants always apply prompt to multiple layers. For example, VPT-deep integrates prompts starting from one layer and extends them to all subsequent layers. The distinct layer-prompting configuration employed by AP stems from its unique research focus, as mentioned in the previous response. That is, the AP’s setting serves as a natural extension of the input-layer VP. In contrast, VPT draws its inspiration from a similar design commonly found in NLP.

* (**Layer Preference Study and Its Model Architecture Effect**) In this work, AP serves as a general notion of VP and a “medium” to reveal the layer preference of vision models on “prompts”. Through AP, we can gain both theoretical and practical insights into these layer preferences across different model types, ascertain the (in)effectiveness of the input-level VP, and eventually narrow the performance gap of VP/AP with other PEFT methods. This contrasts with VPT's approach, which lacks a systematic examination of layer effects and has no in-depth analysis on its architecture effect (i.e., CNNs vs. ViTs). While we acknowledge that VPT has done one ablation study on layer depth to install prompt, the key conclusion on the layer preference as well as its model architecture effects are all absent in the VPT work. Therefore, despite some similarity of the prompting tyle shared by AP and the deep variant of VPT, AP brings its unique insights and contribution on understanding and improving VP.

* (**Theoretical Understanding**) Besides the empirical exploration, one of the key distinctions of our work from the VPT study lies in the comprehensive theoretical analysis we have conducted. This theoretical exploration is twofold. First, we have elucidated a clear and provable link between AP and normalization tuning (see “Understanding AP through its connection to normalization tuning” in Sec. 3, and Proposition 1 in Appendix C.2). This connection is crucial as it underpins the operational mechanism behind AP, offering a theoretical validation of its effectiveness. This aspect of our work not only substantiates the empirical findings but also offers a deeper, rigorous understanding of how AP functions and its implications. Second, another critical theoretical contribution of our research is the validation of the layer preference phenomenon and its impact on different architectures (see Sec. 4.2, Appendix C.3 and C.4). Our theoretical analysis confirms that the input-level VP, as traditionally implemented, is indeed suboptimal. This theoretical support provides a robust foundation for our empirical findings.

**Contributions of this work.** While the previous responses have effectively communicated our contributions, we want to underscore the key ones below.

* **Insights from AP**: Utilizing AP, we uncovered that traditional input-level VP designs are inherently suboptimal. Regarding the strength of AP, we also established a provable relationship between AP and normalization tuning (Section 3) to illustrate its mechanism, as acknowledged by Reviewer @Z13f.

* **Layer Preference Observation and Insights**: We empirically demonstrated the layer preference and its architecture effects. Through empirical studies, we unveiled the connection between the layer preference and the capacity for capturing global features (Section 4.1). Furthermore, we **theoretically** validated those findings in different model architectures (Section 4.2), as recognized by Reviewers @TRtL and @Z13f.

* **Bridging the Performance Gap**: Our extensive experiments highlight that AP consistently outperforms traditional VP, as evident in Tables 1, 2, and 3. Furthermore, it significantly narrows the performance gap between VP and other PEFT methods, as demonstrated in Table 4. It is important to clarify that our aim is not to claim that AP outperforms all PEFT methods. Rather, AP's effectiveness lies in its ability to notably reduce the gap with state-of-the-art PEFT methods by simply prompting the appropriate network layer.

---

### Author Response · Authors · 2023-11-20
**Highlighted General Response (GR) - Part I**

We extend our gratitude to all the reviewers for their invaluable comments. In what follows, we offer two general responses aimed at addressing potential misconceptions about our work. Additionally, we provide a concise summary of the experiments we have incorporated in response to the insightful suggestions made by the reviewers.


## GR 1: Misunderstanding of the contributions of this work and its distinctions with VPT

A few concerns were raised by reviewers about the novelty of this work (@Reviewer Yfja and @Reviewe jt12), particularly regarding the motivation of our proposed AP (activation prompt) and its comparison with  VPT. We recognize these concerns and believe that there might exist  some possible misunderstandings. Therefore, we would like to clarify our contributions and distinguish them from VPT.

**Clarification on the focus of our research** (@Reviewer Yfja and @Reviewer jt12)

In our study, AP is motivated by the input-level VP (visual prompt), which is applied to both CNNs and ViTs, rather than VPT that mainly focuses on ViTs. To be more specific, previous research has highlighted a notable performance drop when using input-level prompts, even in comparison to the simplest linear probing technique for fine-tuning in transfer learning. This inspires us to re-think the “value” of the input-level VP. Given this, we introduce AP as a natural extension of VP, as it involves the direct addition of prompt perturbations to the intermediate “input” (i.e., activation map) at a given layer. Through this approach, we can then gain insights into how the positioning of the prompt within the layer (referred to as "layer preference") influences the prompting performance. Here the input layer can be viewed as the beginning layer. By doing so, we could answer whether the use of input-level prompts is indeed a favorable choice in parameter-efficient fine-tuning (PEFT) for vision tasks.

As explained above, our work aims to understand, advance the input-level VP and reduce its gap with the state-of-the-art (SOTA) PEFT methods. We introduce AP as a tool to gain insights from VP and reveal its intrinsic limitations. In addition, our exploration on the “layer preference” of AP across different model architectures (CNNs vs. ViTs) enriches the existing VP design principles, offering both theoretical and practical insights to VP. Given the above research focus, the SOTA VP method is chosen as the major baseline in this work as detailed in Table 1, Table 2 and Table 3. It's essential to note that our primary objective is **not** to propose a method that can outperform all PEFT methods in every scenario. Instead, we view the reduced gap between AP and existing PEFT methods as a significant positive outcome, showing the progress of advancing the input-level VP.

---

### Meta-Review · Area_Chair_3Yni · 2023-12-11

**Metareview:**

The paper introduces Activation Prompt (AP), as an extension of Visual Prompt Tuning (VPT-deep), aiming to bridge the performance gap between visual prompting and fine-tuning while emphasizing computational efficiency. AP involves perturbing activation maps in internal layers, connecting with normalization tuning. AP consistently outperforms traditional Visual Prompting (VP) and various other parameter-efficient fine-tuning (PEFT) methods, highlighting its efficacy in achieving improved performance compared to existing approaches. The manuscript is praised for its clarity, coherence, and well-motivated presentation, contributing valuable insights into the domain of efficient model adaptation. The reviews acknowledge the paper's clarity, coherence, and extensive experimentation across 29 datasets. However, the final scores are 5,5,6,5, which is a borderline case mainly due to the weakness (below).

The identified weaknesses center on the limited exploration and comparison within the proposed Activation Prompt (AP) method, with critics noting its diminished novelty due to its close relation to Visual Prompt Tuning (VPT-deep). The absence of a comprehensive layer-wise comparison between VPT-deep and AP hinders nuanced performance assessment. The paper lacks a detailed explanation for AP's superiority over VPT-deep and explores alternative feature-wise aggregation strategies insufficiently. Furthermore, the omission of a comparison with state-of-the-art parameter-efficient fine-tuning (PEFT) algorithms limits insights. While some concerns are addressed in the rebuttal, the core question remains: What is AP's unique contribution, being a specific implementation within VPT-deep? Consequently, the meta-reviewer recommends rejection.

**Justification For Why Not Higher Score:**

There are many concerns arise about the method's novelty, specifically its proximity to VPT-deep, and the lack of a detailed explanation for its superior performance. The reviewers suggest layer-wise comparisons, exploration of alternative strategies, and comparisons with state-of-the-art parameter-efficient fine-tuning methods for a more comprehensive evaluation. Some refinements and deeper exploration are

**Justification For Why Not Lower Score:**

N/A

---

### Decision · Program_Chairs · 2024-01-16

Reject